# Learning Actionable Counterfactual Explanations in Large State Spaces

## Abstract

An increasing number of high-stakes domains rely on machine learning to make decisions that have significant consequences for individuals, such as in loan approvals and college admissions. The black-box nature of these processes has led to a growing demand for solutions that make individuals aware of potential ways they could improve their qualifications. Counterfactual explanations (CFEs) are one form of feedback commonly used to provide insight into decision-making systems. Specifically, contemporary CFE generators provide explanations in the form of *low-level* CFEs whose constituent actions precisely describe how much a negatively classified individual should add to or subtract from their input features to achieve the desired positive classification. However, the *low-level* CFE generators have several shortcomings: they are hard to scale, often misaligned with real-world conditions, constrained by information access (e.g., they can not query the classifier), and make inadequate use of available historical data. To address these challenges, we propose three data-driven CFE generators that create generalizable CFEs with desirable characteristics for individuals and decision-makers. Through extensive empirical experiments, we compare the proposed CFE generators with a *low-level* CFE generator on four real-world (BRFSS, Foods, and two NHANES datasets), five semi-synthetic, and five variants of fully-synthetic datasets. Our problem can also be seen as learning an optimal policy in a family of large but deterministic Markov decision processes.

## 1 Introduction

Machine learning models are increasingly used to guide consequential decision-making. Since these decisions can significantly impact livelihoods, society demands the right to explanation, as stated in Articles 13–15 of the European Parliament and Council of the EU (2016) General Data Protection Regulation. One of the most needed explanations is how individuals (agents) can modify their state (i.e., the input features to the models) to achieve a (desirable) positive classification. Counterfactual explanations (CFEs) provide one such solution in the form of actionable insights (Wachter et al., 2017; Dandl et al., 2020; Mothilal et al., 2020; Ustun et al., 2019; Karimi et al., 2021; Joshi et al., 2019; Karimi et al., 2022). Most contemporary CFE generators like actionable recourse (Ustun et al., 2019), provide *low-level* CFEs, where each action specifies the precise amount by which the individual should add to or subtract from a specific feature to ensure that the new features collectively result in a positive classification. For example, if an individual is classified as having an unhealthy waist-to-hip ratio (WHR), one of the recommended low-level actions to help them achieve a healthier WHR, as shown in Figure 1(a) blue, is to "*increase selenium (mg) from* 45 *to* 327.7319."

However, such low-level CFEs exhibit several notable shortcomings (Figure 1) that limit their effectiveness in practice. As we discuss in Section 2, these include a focus on precise changes to individual features, which can make them difficult for a person to act upon; high computational complexity that affects scalability; a reliance on access to potentially privileged information (e.g., the ability to query the classifier); and a limited ability to utilize existing domain knowledge or historical data. To address these limitations, we propose three novel data-driven CFE generator frameworks: *hl-continuous* (high-level continuous), *hl-discrete* (high-level discrete), and *hl-id* (high-level identifier) CFE generators (see Section 3). Each proposed CFE generator produces generalizable CFEs that empower individuals to use their agency to gain capabilities that favorably transform their current state (features).

| List of actionable features | Input state (individual's nutrient intake) $\mathbf{x}$ | The *hl-continuous* CFE (2 actions) | | The *low-level* CFE (19 actions) Format of *low-level* actions: ↑ or ↓ the feature from − to (→) − |
|---|---|---|---|---|
| | | **action-1:** *"take leavening agents: cream of tartar"* | **action-2:** *"take fish, tuna, light canned in water drained solids"* | |
| Calcium (mg) | 309 | 8.000 | 17.000 | 309 → 113 |
| Carbohydrate (gm) | 109.45 | 61.500 | 0.000 | 109.45 → 43.37600000000005 |
| Copper (mg) | 0.425 | 0.195 | 0.050 | 0.425 → 0.2129500000000001 |
| Dietary fiber (gm) | 4.1 | 0.200 | 0.000 | 4.1 → 50.113950000000024 |
| Iron (mg) | 4.08 | 3.720 | 1.630 | 4.08 → 42.72946000000002 |
| Magnesium (mg) | 96 | 2.000 | 23.000 | 96 → 57 |
| Niacin (mg) | 8.755 | 0.000 | 10.136 | 8.755 → 85.10721500000001 |
| Phosphorus (mg) | 488 | 5.000 | 139.000 | 488 → 217 |
| Potassium (mg) | 994 | 16500.000 | 179.000 | 994 → 6520.550000000004 |
| Protein (gm) | 29.03 | 0.000 | 19.440 | |
| Selenium (mcg) | 45 | 0.200 | 70.600 | 45 → 327.7319 |
| Sodium (mg) | 1326 | 52.000 | 247.000 | 1326 → 626.65 |
| Total folate (mcg) | 172 | 0.000 | 4.000 | 172 → 1179.7380000000003 |
| Total monounsaturated fatty acids (gm) | 12.392 | 0.000 | 0.107 | 12.392 → 89.34236700000017 |
| Total polyunsaturated fatty acids (gm) | 13.999 | 0.000 | 0.277 | 13.999 → 4.40896 |
| Total saturated fatty acids (gm) | 10.077 | 0.000 | 0.211 | 10.077 → 2.6004500000000004 |
| Vitamin B6 (mg) | 0.482 | 0.000 | 0.319 | 0.482 → 0.21794999999999998 |
| Vitamin B12 (mcg) | 1.21 | 0.000 | 2.550 | 1.21 → 0.12000000000000001 |
| Vitamin C (mg) | 35.7 | 0.000 | 0.000 | 35.7 → 0.1000000000000142 |
| Zinc (mg) | 2.61 | 0.420 | 0.690 | 2.61 → 1.3895 |

(a) For an individual with input features $\mathbf{x}$ yellow and a negative WHR classification, the low-level CFE blue suggests 19 actions, while the hl-continuous CFE orange suggests 2.

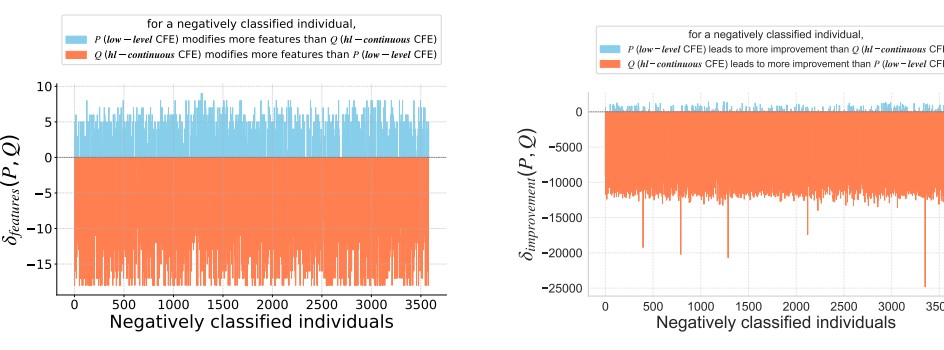

(b) Difference in number of modified features

(c) Difference in improvement achieved

Figure 1: For an individual negatively classified as having an unhealthy WHR (a)(yellow), to help them make changes that lead to a healthy WHR classification, the low-level CFE generator suggests a unique CFE (a)(blue) with 19 actions, modifying 19 features at the cost of 56.588, resulting in an improvement of 5679.95. In contrast, the hl-continuous CFE generator recommends a CFE (a)(orange) with only two actions—"*take leavening agents: cream of tartar*" and "*take fish, tuna, light, canned in water, drained solids*"—at a cost of 4.010, modifying 19 features but achieving a higher improvement of 16682.62, and the CFE optimal for 105 other agents. An investigation of the difference in number of modified features ($\delta_{features}(P, Q)$) and difference in improvement achieved ($\delta_{improvement}(P, Q)$) when each negatively classified WHR agent takes a low-level CFE (P) vs. an hl-continuous CFE (Q), shows that hl-continuous CFEs modify more features (b) and lead to significantly higher improvement (c) than low-level CFEs.

The effects of the CFE's actions on an individual's state are explicit in hl-continuous and hl-discrete CFEs, but not in an hl-id CFE. Specifically, an hl-continuous action is a signed ($\pm$) and named, general predefined action that might modify several features simultaneously (e.g., **action-1** (orange) in Figure 1(a) simultaneously modifies 11 features). An hl-continuous CFE is then the lowest-cost set of hl-continuous actions, which is solution of a integer linear program (ILP). That is, given a negatively classified individual and a set of hl-continuous actions with known costs, the goal of the ILP is to find the lowest-cost subset that modifies the individual's features to achieve a positive classification. We propose a deep learning-based hl-continuous CFE generator that, given instances of individuals and their corresponding hl-continuous CFEs, can quickly and accurately generate hl-continuous CFEs for new individuals without generator re-optimization.

On the other hand, an hl-discrete action is a binary action that specifies whether an action fulfills the required capabilities for a specific feature. This formulation of actions is particularly efficient in scenarios where each feature's satisfiability is based on the feature's respective threshold and can be reduced to a yes/no question. For example, in level one decision-making, e.g., wellness,

customer satisfaction, and compliance checks, an individual must satisfy a subset of prerequisites to guide subsequent decisions. We formulate the hl-discrete CFE as a solution to a weighted set cover problem. Specifically, given a set of hl-discrete actions with known costs and effects on binary state features, the problem is to find the lowest-cost subset that modify the individual's state such that they become positively classified. We propose a deep learning hl-discrete CFE generator trained on instances of individuals and their optimal hl-discrete CFEs (individual↦hl-discrete CFE dataset) to generate hl-discrete CFEs for new individuals.

Lastly, an hl-id CFE is a unique identifier (or name) for a CFE. It is particularly efficient for settings where decision-makers have minimal information access, for example, no query access to the classifier, and the actions and their costs and explicit effects on the features are unknown. It is also often the case that the hl-id CFE holds significant implicit information. For instance, a registered dietitian might recommend the hl-id CFE, "*remove gluten from the child's diet*" to a parent of a child diagnosed with celiac disease to flip the diagnosis. The dietitian generates this CFE based on historical patient-CFE (intervention) information, even without direct access to the celiac classifier and without specifying a comprehensive list of restricted foods and their effects on relevant features. More detailed examples are provided in Section 3.3.

## 2 BACKGROUND

We consider a binary classification setting, where an individual with state $\mathbf{x}$ receives either a positive (desirable) or negative (undesirable) classification under a model $f(\mathbf{x})$. Although we focus on this setting, our proposed CFE generation framework generalizes to other scenarios. Given an individual state $\mathbf{x}$ with an undesirable model outcome, the objective of the CFE generator is to provide the individual with information that they can act on to achieve a desirable classification under the model. Contemporary low-level CFE generators, such as actionable recourse (Ustun et al., 2019), provide low-level CFEs where each action in the CFE precisely specifies how much the individual should add or subtract from a specific feature to ensure that, collectively, the new features (state) result in the individual receiving a desirable model outcome.

**The low-level CFE generator** Ustun et al. (2019) proposed an ILP-based low-level CFE generator (Equation 1) that generates a low-level CFE to help an individual change an undesirable model outcome to a desirable one.

$$
\begin{aligned}
\min \quad & \text{cost}(\mathbf{a}; \mathbf{x}) \\
\text{s.t.} \quad & f(\mathbf{x} + \mathbf{a}) = \hat{y}^\star \\
& \mathbf{a} \in A(\mathbf{x}),
\end{aligned} \tag{1}
$$

where $\hat{y}^\star$ is the desired model outcome, $A(\mathbf{x})$ denotes the set of feasible actions given the input $\mathbf{x}$, and the function $\text{cost}(\cdot; \mathbf{x}) : A(\mathbf{x}) \to \mathbb{R}_+$ encodes the preferences between these actions. When Equation 1 is feasible, the optimal actions that modify the features (i.e., $\mathbf{x} + \mathbf{a}$) and lead to a desirable model outcome are recommended to the individual (Figure 1(a) blue). We refer the reader to Ustun et al. (2019) for a more detailed description and to Appendix C.1 for dataset-specific experimental setup and supplemental examples of this low-level CFE generator.

**Shortcomings of low-level CFE generators** We address four notable limitations of the low-level CFE generators (Verma et al., 2020; Karimi et al., 2022; Barocas et al., 2020). First, they are hard to scale due to the need to solve a computationally intensive NP-hard optimization problem for each new agent (Karp, 1972; Karimi et al., 2022), and the CFE's actions are overly specific (e.g., Figure 1(a) blue). Second, some assumptions about the problem structure may not hold in the real world. For instance, most assume that the CFE's actions are in final implementable steps and that each action directly modifies an individual feature, thus the need for high sparsity (few modified features) and high proximity (minimal improvement). Third, if there are information access challenges, i.e., no access to critical information—such as the classifier data and parameters, a prediction training data manifold to ensure diverse, representative and optimal CFEs, or a complete list of actions and their costs—contemporary CFE generation becomes infeasible, biased, or flawed. Lastly, in real-world contexts, there might be data on historical mappings of individuals and their CFEs that the contemporary CFE generation does not adequately leverage, limiting its effectiveness.

# 3  DATA-DRIVEN CFE GENERATION

We propose three data-driven CFE generators of increasing generality, hl-continuous, hl-discrete, and hl-id. The proposed CFE generators work under various information access constraints, leverage data beyond that key to classification (e.g., classifier parameters and predictive training data) and generalize to negatively classified individuals beyond those on which the model was trained.

## 3.1  THE HL-CONTINUOUS CFE GENERATORS

The data-driven hl-continuous CFE generator is trained on instances of individual states paired with their corresponding hl-continuous CFEs to generate respective CFEs for new individuals without generator re-optimization . Our empirical results demonstrate that even a simple deep-learning based hl-continuous CFE generator performs strongly at this task. In the following, we provide formal definitions of hl-continuous actions and hl-continuous CFEs, while Section 4.2 includes detailed descriptions of the experimental generator model architecture.

**Definition 1.** (hl-continuous action) : An hl-continuous action is a signed ($\pm$) and named, general predefined action whose cost and varied effects on an individual's input features are predefined and known. For example, **action-1**(orange) in Figure 1(a), "*take leavening agents: cream of tartar*" adds nutritional values to 11 nutrients by a known amount and incurs a cost (e.g., estimated average price in USD) that is known a priori.

**Definition 2.** (hl-continuous CFE): An hl-continuous CFE is a solution to an ILP where, given a negatively classified individual state $\mathbf{x}$ and a set of hl-continuous actions with known costs, the problem is to find the lowest-cost subset of hl-continuous actions that when taken, can favorably modify the individual's state. The ILP is of the form:

$$
\begin{aligned}
\text{minimize} \quad & \sum_{j \in J} \text{cost}_j a_j \\
\text{s.t.} \quad & \mathbf{c}^T \sum_{j \in J} a_j \cdot (2\epsilon_j - 1) \cdot \mathbf{v}_j \geq -(\mathbf{c}^T \mathbf{x} + b) + \delta \\
& \epsilon_j \in \{0, 1\}, \quad a_j \in \{0, 1\}, \quad \forall j \in J
\end{aligned}
\tag{2}
$$

where $J$ denotes the indices of the hl-continuous actions, with each action represented by a vector $\mathbf{v}_j$ and with a predefined cost, $\text{cost}_j \in \mathbb{R}_+$. The boolean variable $a_j$ indicates the inclusion ($a_j = 1$) or exclusion ($a_j = 0$) of the $j^{\text{th}}$ hl-continuous action, while $\epsilon_j$ encodes the sign of this action, representing addition ($\epsilon_j = 1$) or subtraction ($\epsilon_j = 0$). The coefficients $\mathbf{c}$ and intercept $b$ are predefined parameters of the linear classifier, and $\delta$ is a small positive value that ensures strict inequality.

## 3.2  THE HL-DISCRETE CFE GENERATORS

We propose the data-driven hl-discrete CFE generator, trained on individual↦hl-discrete CFE data, to quickly and accurately produce hl-discrete CFEs for new individuals. Below, we formally define hl-discrete actions and hl-discrete CFEs and defer further details about the experimental model architecture of the hl-discrete CFE generator to Section 4.2.

**Definition 3.** (hl-discrete action): An hl-discrete action is a binary vector that specifies which features the action adds capabilities. For example, consider the individual state $\mathbf{x} = [0, 0, 0, 0, 1]$ and the hl-discrete action $\mathbf{v}_j = [1, 1, 0, 0, 0]$. When taken, the hl-discrete action adds capabilities to features 1 and 2 of $\mathbf{x}$, transforming it to a new state $[1, 1, 0, 0, 1]$. Although we focus on binary actions, the setting can be extended to more general cases.

**Definition 4.** (hl-discrete CFE): An hl-discrete CFE is formulated as a solution to a weighted set cover problem. Specifically, the CFE is the lowest-cost subset of hl-discrete actions, each with predefined costs, that a negatively classified individual $\mathbf{x} \in \{0, 1\}^n$ (e.g., someone deemed a health risk) can undertake to achieve a desirable classification (e.g., no longer classified as a health risk).

The problem can be formally defined as follows:

$$
\begin{aligned}
\text{minimize} \quad & \sum_{j \in J} \text{cost}_j a_j \\
\text{s.t.} \quad & \sum_{j \in J} d_{ji} a_j + x_i \geq t_i, \ \forall i \in [n], \\
& a_j \in \{0,1\}, \ d_{ji} \in \{0,1\},
\end{aligned}
\tag{3}
$$

where $J$ are the indices of the hl-discrete actions, each represented by a vector $\mathbf{v}_j$ and with a predefined cost: $\text{cost}_j \in \mathbb{R}_+$. The threshold classifier $\mathbf{t} = \{t_1, t_2, \cdots, t_n\}$ over $n$ features classifies an individual state $\mathbf{x}$ positive if $x_i \geq t_i$, $\forall i \in [n]$, and negative otherwise. The binary variable $a_j$ denotes inclusion ($a_j = 1$) or exclusion ($a_j = 0$) of the $j^{\text{th}}$ hl-discrete action, while $d_{ji}$ indicates whether the $j^{\text{th}}$ hl-discrete action transforms (adds capabilities to) the feature $i$ of the individual state $\mathbf{x}$, i.e., when performed, the new individual state $\mathbf{x} + \mathbf{v}_j = \mathbf{x}'$ is such that $x'_i > x_i$ and $x'_i \geq t_i$.

### 3.3 THE HL-ID CFE GENERATORS

The hl-id CFE generator is a supervised learning model trained on an individual$\mapsto$hl-id CFE dataset to generate hl-id CFEs (unique CFE identifiers) for new individuals. Details on the experimental model architecture are provided in Section 4.2. Typically, detailed information about the actions within each hl-id CFE—including the costs and the specific effects of the actions on input features—is unknown, and decision-makers cannot query the classifier. This approach is instrumental when the CFE unique identifier conveys significant implicit information. For example, consider two health-related scenarios: 1) an individual diagnosed with an unhealthy heart condition could receive an hl-id CFE such as "*cardiac rehabilitation*" (Fernández-Rubio et al., 2022), without direct access to the heart diagnostic classifier or specifying underlying actions (e.g., aerobics exercises); 2) an individual classified with an unhealthy weight might be assigned an hl-id CFE such as "*adopt a ketogenic diet*," without query access to the classifier or specifying sub-actions involved or which nutrients they change and by how much (e.g., add leavening agents: cream of tartar to their diet).

## 4 EXPERIMENTAL SETUP

We empirically evaluate the three proposed data-driven CFE generators against the low-level generator using various metrics (see Appendix A). For example, we use $\delta_{features}(P, Q) = |P_{\text{features}}| - |Q_{\text{features}}|$ to measure the difference in the number of modified features when an individual takes CFE $P$ vs. $Q$. To assess accuracy of the proposed generators, we use zero-one loss (see Equation 4), which checks if the generated CFE $\hat{I}$ matches the true CFE $I$.

$$
\mathcal{L}_{\text{eval}}(I, \hat{I}) = \begin{cases} 0 & \text{if } I = \hat{I} \\ 1 & \text{if } I \neq \hat{I} \end{cases}
\tag{4}
$$

### 4.1 DATASETS

We conducted experiments with 4 real-world, 5 semi-synthetic, and 5 variants of fully-synthetic datasets. Each of the individual$\mapsto$CFE datasets (instances of individuals and their corresponding CFEs) was split $80/20$ for training and evaluation of data-driven CFE generators. While generalizable to other cases, we focused on a setting where each individual in the respective individual$\mapsto$CFE datasets has one optimal CFE match, categorized as either hl-continuous, hl-discrete, or hl-id, depending on the dataset considered. The following provides key details about the datasets used in the experiments. Further information, including the preprocessing procedure and the specific nature of the feature representations, is available in Appendices B.1 and B.2.

**The real-world datasets** We use four real-world datasets. The first is the Behavioral Risk Factor Surveillance System (BRFSS) dataset (Teboul, 2024b; Centers for Disease Control and Prevention, 2024), consisting of 23617 individuals with 16 binary health risk factors after preprocessing.

Additionally, we extracted the BMI (body mass index) and WHR (waist-to-hip ratio) datasets from NHANES body measurement surveys (CDC, 1999; ICPSR at the University of Michigan, 2024)

for the years 1999 to pre-pandemic 2020. After preprocessing, the BMI dataset contained 50918 individuals, each with 3 demographic and 19 nutrient intake features, and classified as healthy (1) or unhealthy (0) BMI. The WHR dataset contained 9120 individuals, each with 3 demographic and 20 nutrient intake features, and classified as either healthy (1) or unhealthy (0) WHR.

After preprocessing, the extracted Foods dataset contains 3901 food items, each with details on portions and nutritional compositions (USDA, Agricultural Research Service, Nutrient Data Laboratory, 2016; Awram, 2024). For each food item, we add two types of costs: *monetary cost* in USD (obtained via internet scraping) and *caloric cost*, reflecting each food's caloric content (Caputo, 2023).

**The semi-synthetic datasets** Using the BMI and WHR datasets and the two types of hl-continuous actions defined by the Foods dataset, with each food having costs defined by either monetary or caloric costs, i.e., Foods+monetary costs and Foods+caloric costs (refer to Appendix B.1 for further details), we created four individual↦hl-continuous CFE datasets using ILP (Equation 2) and dataset-specific hyperparameter-tuned logistic regression models. Additionally, using the BRFSS dataset and the ILP defined in Equation 3 with a threshold classifier $\mathbf{t} = \mathbf{1}_n$, we generated a semi-synthetic individual↦hl-discrete CFE dataset.

We used the unique identifiers for the CFEs to create three individual↦hl-id CFE datasets from the following individual↦CFE datasets: the BMI dataset with Foods+monetary cost actions, the WHR dataset with Foods+caloric cost actions, and the individual↦hl-discrete CFE BRFSS dataset.

Lastly, for each of the semi-synthetic individual↦CFE datasets described above, before the train/test split, we generated three "*varied frequency of CFEs*" datasets: `all` (including all data), `>10` (more than 10 individuals per CFE), and `>40` (more than 40 individuals per CFE).

**The fully-synthetic datasets** We use the ILP defined in Equation 3 to generate five variants of the individual↦hl-discrete CFE datasets: varied dimensionality, frequency of CFEs, information access, feature satisifiability, and actions access. Below, we briefly describe some of the variants and include more details about these and other variants in Appendix B.2.

For "*varied dimensionality*", we generated datasets with 20, 50, and 100 dimensions (actionable features), where we set the individual's feature to 1 with a probability $p_f$, and each *discrete action* can add capabilities to a feature with a probability $p_a$. The cost of each action depends on the features it transforms. Lastly, we created three varied frequency of CFEs datasets—`all`, `>10`, and `>40`—, individual↦hl-id CFE datasets for each varied dimensionality dataset, using a similar approach as in the semi-synthetic individual↦CFE datasets described above.

## 4.2 CFE GENERATOR ARCHITECTURES

Below, we provide important details about the experimental model architectures for the data-driven CFE generators, with more information included in Appendix C.2.

**The hl-continuous CFE generator model** Although generalizable to other settings, we use the names and costs of the hl-continuous actions of the CFEs in the individual↦hl-continuous CFE dataset, e.g., {action-a, action-b, and action-c} and their corresponding costs: {cost-a cost-b, and cost-c} to design the generator model. We design the model as a neural network with three hidden layers, each with 2000 neurons, $\ell_2$ regularization, dropout, and batch normalization. We used the Adam optimizer (Kingma & Ba, 2014) and implemented early stopping with the best weights restored after a patience level of 300. We set the batch size to 6000 and the number of epochs to 5000, on average. To ensure that the hl-continuous CFE generator performs well on the training individual↦CFE dataset and accurately generates hl-continuous CFEs for new individuals, we optimize the model loss function $\mathcal{L}_{\text{FA}}$ given by:

$$\mathcal{L}_{\text{FA}} = -\frac{1}{M} \sum_{m=1}^{M} \sum_{j=1}^{J} [a_{jm} \log(\hat{a}_{jm}) + (1 - a_{jm}) \log(1 - \hat{a}_{jm})] \tag{5}$$

where $\hat{a}_{jm}$ is the predicted probability and $a_{jm}$ is the true indication of a presence (1) or absence (0) of the $j^{\text{th}}$ hl-continuous action in individual $m$'s hl-continuous CFE. There are $J$ possible hl-continuous actions and $M$ individuals in the individual↦hl-continuous CFE training dataset.

**The hl-discrete CFE generator model**  We design a sequential encoder-decoder network to generate hl-discrete CFEs for new individuals. The model is trained on a dataset comprising instances of individuals and their associated hl-discrete CFEs, enabling it to quickly and accurately predict CFEs for previously unseen individuals. The model configuration varied depending on the experimental setting. On average, we used 500 training epochs with a batch size of 128, a dropout rate of 0.4, a learning rate of 0.0005, and either the mean squared error loss or binary cross-entropy loss as the objective function. The encoder and decoder networks typically consisted of three layers, each using ReLU activation functions.

**The hl-id CFE generator model**  Given the individual↦hl-id CFEs training dataset, we design a neural network model with an average of two hidden layers, each consisting of 2000 neurons, $\ell_2$ regularization, dropout, and batch normalization. We used the Adam optimizer (Kingma & Ba, 2014) and implemented early stopping and restoration of the best weights after a patience level of 360. On average, we set the batch size to 2000 and the number of epochs set to 3000. To ensure that the hl-id CFE generator performs well on the training dataset and accurately generates hl-id CFEs for new individuals, we optimize the model loss function $\mathcal{L}_{\text{NC}}$ given by:

$$\mathcal{L}_{\text{NC}} = -\frac{1}{M} \sum_{m=1}^{M} \sum_{k=1}^{K} [a_{km} \log(\hat{a}_{km})] \tag{6}$$

where $\hat{a}_{km}$ is the predicted probability and $a_{km}$ is the true indication of the $k^{\text{th}}$ CFE being the hl-id CFE (1) or not (0) for the $m^{\text{th}}$ individual. There are $K$ possible hl-id CFEs and $M$ individuals in the training dataset.

## 5 EXPERIMENTAL RESULTS

In this section, we provide thorough empirical evidence to show the strong performance of our generators and how and in what ways in comparison to low-level CFEs, hl-continuous, hl-discrete and hl-id CFEs, might be preferable to both individuals and decision-makers.

### 5.1 THE HL-DISCRETE AND HL-CONTINUOUS CFES ARE PREFERABLE

Below and in Appendices D.1 and D.2, we provide empirical evidence to show that, compared to low-level CFEs, both hl-continuous and hl-discrete CFEs involve fewer actions, lead to more diverse improvements, are easier to personalize, and simplify the design and interrogation of CFE generators for fairness issues. Additionally, they more accurately reflect real-world conditions.

**Sparsity**  In low-level CFE generation, sparsity—typically defined as a small number of modified features (Verma et al., 2020)—is often a primary goal due to the presumed one-to-one relationship between number of actions taken and features modified. However, achieving sparsity in practice may be both undesirable and challenging because individuals often aim to implement as many changes as possible with minimal actions, and is rarely the case that each action modifies one feature (see Figure 2(a)). Our results, as shown in Appendix D.1 and demonstrated here for the WHR dataset with Foods+caloric cost actions, underscore this point.

For example, the hl-continuous actions in Figure 1(a) orange modify several features simultaneously. Additionally, while in low-level CFEs there is a perfect positive correlation between the number of modified features and actions (Kendall's $\tau = 1.0$, *p-value* $= 0.0$), the correlation between the number of modified features and actions in hl-continuous CFEs is positive but not perfect (Kendall's $\tau = 0.722$, *p-value* $= 0.0$). Furthermore, as the number of modified features decreases in low-level CFEs, a different trend is observed for hl-continuous CFEs (Kendall's $\tau = -0.233$, *p-value* $= 1.7e$-73). Lastly, despite hl-continuous CFEs having fewer actions on average ($\sim 2$), they result in more feature changes ($\sim 16$) compared to low-level CFEs, which have an average of $\sim 9$ actions and $\sim 9$ feature changes (refer to Figures 1 and 2(a)).

**Proximity**  Similar to sparsity, maximizing proximity—defined as ensuring that the new state after taking the CFE is close to the initial state (Verma et al., 2020)—is based on the real-world assumption of a strong positive correlation between proximity and the number of actions taken. However,

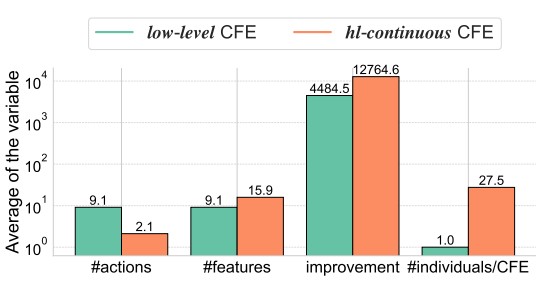 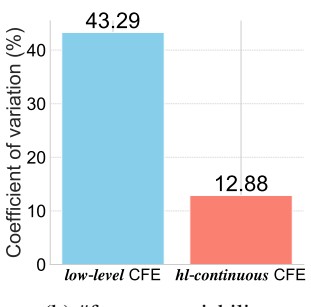

(a) Average of the variables          (b) #features variability

Figure 2: On average, (a) hl-continuous CFEs involve fewer actions, modify more features, lead to states more distant from current states (higher improvement), and have a higher CFEs frequency (average of number of individuals per CFE is 27.5) than low-level CFEs. Additionally, (b) there is more variability in number of modified features across sensitive groups (less fairness) with low-level CFEs than hl-continuous CFEs (see Figures 11 and 12 and Appendix D.2 for more details).

while high proximity suggests fewer changes, it also implies minimal improvement (i.e., a small distance between the initial and new state), which can be undesirable and challenging to achieve in practical settings. Our results, as demonstrated in Appendix D.1 and here with the WHR dataset and Foods+caloric cost actions, show that hl-continuous CFEs typically involve fewer actions but lead to more distant states – higher improvement (see Figures 1 and 2(a)).

Although there is a significant positive correlation between improvement in hl-continuous CFEs and improvement in low-level CFEs (Kendall $\tau = 0.542$, *p-value* = 0.0), there is almost no relationship between the number of actions taken and improvement achieved in hl-continuous CFEs (Kendall $\tau = 0.0625, p\text{-}value = 3.21e\text{-}06$). In contrast, there is a notable positive correlation between actions taken and improvement achieved in low-level CFEs (Kendall $\tau = 0.368, p\text{-}value = 5.41e\text{-}227$. Additionally, on average, hl-continuous CFEs, with fewer actions ($\sim 2$) result in states that are more distant (improvement: $\sim 12765$) compared to low-level CFEs, which typically involved $\sim 9$ actions and achieved an improvement of $\sim 4485$ (see Figure 2(a)).

**Diverse and higher improvement**  A key observation from the differences in sparsity and proximity between low-level CFEs and hl-continuous or hl-discrete CFEs, as described earlier, is that hl-continuous CFEs tend to be more desirable for decision-makers and individuals alike. For decision-makers, these CFEs make individuals more "positive" or "qualified." For individuals, hl-continuous CFEs are preferable because they involve fewer, more clearly defined actions, lead to more diverse and substantial improvements (modify more features and result in distant states from the current state)), and reduce the costs associated with interpreting and executing the CFEs (see Figures 1 and 2(a) and in Appendix Figures 8 and 10).

**Personalization and fairness**  Since both hl-discrete actions and hl-continuous actions are predefined and general, it is more straightforward and transparent to examine the data-driven hl-discrete and hl-continuous CFE generators for potential fairness issues and to tailor the generation of CFEs to individual needs. For example, our hl-continuous CFE generators can produce CFEs for individuals who place greater importance on monetary costs over caloric costs. Moreover, in general, the hl-continuous and hl-discrete CFEs have less variability in number of actions taken and modified features (e.g., Figure 2(b)) and improvement achieved by individuals across various sensitive groups (more fair), in comparison to low-level CFEs (see Appendices D.2.1 and D.2.2). Lastly, when there are restrictions on the actions individuals have access to or variations in feature satisfiability, our hl-discrete CFE generators demonstrate strong performance in generating CFEs for diverse individuals, even without explicit knowledge of grouped actions or varied feature thresholds (see Appendices D.2.3 and D.2.4).

## 5.2 THE CFE GENERATORS ARE ACCURATE, MORE RESOURCE-EFFICIENT AND SCALABLE

Our results demonstrate that the proposed data-driven CFE generators, operating under various information access constraints—such as no query access to the classifier or, in the case of the hl-id CFE

| | Accuracy of CFE generators | | | | Effect of frequency of CFEs | | |
| --- | --- | --- | --- | --- | --- | --- | --- |
| | hl-continuous | hl-discrete | hl-id | | all | >10 | >40 |
| BMI | $0.92 \pm 0.0053$ | | $0.94 \pm 0.0045$ | 20-dim | $0.84 \pm 0.0060$ | $0.89 \pm 0.0052$ | $0.94 \pm 0.0042$ |
| WHR | $0.92 \pm 0.0176$ | | $0.97 \pm 0.0107$ | 20-dim⋆ | $0.97 \pm 0.0028$ | $0.98 \pm 0.0021$ | $0.99 \pm 0.0014$ |
| BRFSS | | $0.98 \pm 0.0102$ | $0.99 \pm 0.0050$ | BMI | $0.90 \pm 0.0057$ | $0.91 \pm 0.0055$ | $0.92 \pm 0.0053$ |
| 20-dim | | $0.94 \pm 0.0042$ | $0.99 \pm 0.0014$ | BRFSS | $0.70 \pm 0.0182$ | $0.86 \pm 0.0158$ | $0.98 \pm 0.0102$ |

Table 1: (left) The accuracy of the hl-continuous, hl-discrete, and hl-id CFE generators on the new negatively classified individuals for >40, BMI, BRFSS, WHR, and fully-synthetic (20-dim): 20-dimensional, datasets. (right) The CFE generator accuracy decreases with a decrease in the frequency of CFEs in the training set, regardless of the dataset type. Specifically, training and testing on the (20-dim): the 20-dimensional individual↦hl-discrete CFE dataset, (20-dim)⋆: the 20-dimensional individual↦hl-id dataset, (BMI): the BMI individual↦hl-continuous CFE dataset, and (BRFSS): the BRFSS individual↦hl-discrete CFE dataset all show this trend. Our results show that accuracy improves as the frequency of CFEs increases, with generators trained on datasets containing the highest CFE frequency (>40) performing best.

generator, without knowledge of the cost and impact of actions on individual states—are scalable and accurately and efficiently produce CFEs, without requiring re-optimization of the generator (see Table 1(*left*) and Appendices D.3 to D.5). In contrast to the overly specific low-level CFEs, which are generally unique to each individual, hl-continuous and hl-discrete CFEs are often optimal for a broad range of individuals (refer to Figures 1 and 2(a) and Appendix Figure 16). The removal of the need for re-optimization for each new individual, combined with the general applicability of the actions to individuals, enhances the scalability of our proposed CFE generators compared to low-level generators. Additionally, because the actions in the hl-continuous and hl-discrete CFEs are both general and predefined, they are more transparent and easier to interpret (see Figure 1), making them cheaper and more desirable than the overly specific and unique low-level CFEs.

Our results show that the accuracy of the proposed data-driven generators declines with the low frequency of CFEs (see Table 1 (*right*)) and the scalability of CFE generation decreases with an increase in the number of actionable features. We observed that this is due to the growing uniqueness of CFEs (see Table 1 (*right*)) and Appendix D.6). Data augmentation mitigates the negative effects of low CFE frequency. For instance, on the all 20-dimensional dataset, data augmentation improves accuracy from 0.969 to 0.982. Lastly, our proposed data-driven CFE generator performance improves with the complexity of the generator models. For instance, given a discrete, individual↦hl-id dataset, the neural network model outperforms the Hamming distance method (see Appendix Figure 18). Valuable for future works is an exploration of more advanced, data-driven models for CFE generation and techniques like federated learning to facilitate CFE generation under varied data access and privacy constraints.

## 6 LIMITATIONS AND ETHICAL CONSIDERATIONS

The decision-maker must have access to data on instances of individuals and their corresponding optimal CFEs to train the proposed data-driven CFE generators. Although this level of access mitigates some information access challenges—such as needing at least query access to the classifier and representative prediction training data or having an exhaustive list of actions and the associated costs—obtaining historical individual↦CFE data may still pose significant challenges. Future research could investigate techniques like federated learning and secure multi-party computation to collaboratively train robust CFE generators under varied privacy and data access constraints.

Our formulations of hl-continuous and hl-discrete CFEs restrict them to being defined as a set of actions. More generally, one could consider settings where the order of actions matters, such as where a CFE corresponds to an optimal policy for an agent in a deterministic Markov decision process (MDP). Even more generally, one could consider actions whose effects are stochastic, and a CFE then corresponds to an optimal policy for the agent in a general MDP.

The proposed approaches to CFE generation are closely related to data-driven algorithm design. As a result, ethical concerns related to data-driven algorithms, for example, potentially propagating

and exacerbating biases in historical individual↦CFE data and the potential for flawed resource allocation, might apply to our proposed CFE generators. Future research should investigate these ethical implications in greater depth.

Although we focus on health datasets in our experiments, our approach generalizes to a broad spectrum of real-world scenarios, such as college admissions, loan applications, judicial systems, and other settings. Future works could expand our setup to other data settings and informational access challenges. Lastly, we caution readers that the experimentally generated CFEs from our empirical analyses are intended solely for illustrative purposes, and readers should not use them for self-treatment.

## 7 RELATED WORK

Our formulations for hl-continuous and hl-discrete CFEs as solutions to ILPs are in principle, similar to search-based optimization CFE generation frameworks (Ramakrishnan et al., 2019), user-specific ILP recourse approaches (Ustun et al., 2019; Cui et al., 2015; Gupta et al., 2019a), and CFE generation methods based on logic and answer-set programming (Bertossi, 2020; Liu & Lorini, 2023; Marques-Silva, 2023). However, unlike these formulations, we focus on general, predefined actions that often modify multiple features simultaneously (see Figure 1), which could lead to more improvement and help enhance the generalization of the CFE generation.

In addition, contemporary *low-level* CFE generators are often computationally expensive, requiring the solution of NP-hard optimization problems for each new individual. In contrast, we introduce novel data-driven CFE generators that address the question: *Can we, by learning from training data (i.e., instances of individuals and their optimal CFEs), develop a CFE generator that quickly provides optimal CFEs for new individuals?* While in some ways, similar to reinforcement learning-based CFE generation tools (De Toni et al., 2023; Shavit & Moses, 2019; Naumann & Ntoutsi, 2021), our proposed generators offer a more efficient, exact, and scalable alternative to their often high computational and approximate solutions. Notably, our approach is closest to that of Verma et al. (2022). While our method is akin to learning an optimal policy in a large but deterministic family of Markov decision processes (MDPs), Verma et al. (2022) focuses on learning optimal policies within smaller, stochastic MDP settings.

Finally, our work also relates to data-driven algorithm design (Gupta & Roughgarden, 2016; Balcan et al., 2018; Balcan, 2020), where models trained on training data instances perform well on the training data and generalize to the testing data. Unlike contemporary CFE generators that rely solely on classification data (i.e., prediction training data and classifier parameters), our data-driven CFE generators leverage access to individuals and their optimal CFEs and more closely mirror real-world scenarios. Our generators also excel in generating CFEs for new individuals, are more computationally efficient and scalable, and function under varied informational settings. For example, unlike other methods that require, at a minimum, query access to the classifier and knowledge of the cost and impact of each action on state features (Naumann & Ntoutsi, 2021; De Toni et al., 2023; Shavit & Moses, 2019; Verma et al., 2022), our CFE generators—such as the hl-id CFE generator—can effectively produce CFEs without explicit access to any of this information.

## 8 CONCLUSION

In this work, we make a strong case for expanding the focus beyond just classification data (e.g., classifier parameters and prediction training datasets), when automating CFE-based recourse generation. Our findings show that it's more efficient to examine, compare, and personalize the general predefined actions (e.g., hl-continuous and hl-discrete actions), and they significantly enhance the scalability of CFE generation. Additionally, the respective CFEs, hl-continuous, hl-discrete and hl-id CFEs, compared to low-level CFEs are, in retrospect, simpler and more efficient for individuals to execute while yielding more favorable outcomes for decision-makers. Through extensive empirical analysis, we show that the proposed data-driven CFE generators are more scalable, computationally efficient, and better aligned with real-world conditions, all while effectively leveraging data beyond that specific to classification. Our code is available at this: anonymized link.

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

# A  CFE EVALUATION METRICS: SUPPLEMENTAL DETAILS

We compute the difference in number of actions taken, number of modified features, and improvement achieved if an individual took two different CFEs: a low-level CFE and another type, such as an hl-continuous CFE. Specifically, these differences are computed for each training set negatively classified individual.

To compare the low-level CFE to other CFEs, e.g., to analyze sparsity, improvement, fairness, and scalability, we focus exclusively on individual↦CFE datasets derived from computing respective CFEs for negatively classified individuals from the training sets of three datasets: BMI, WHR, and BRFSS. Additionally, since the low-level CFE generator (Equation 1) occasionally fails to generate a CFE for a given individual, we take steps to ensure we more accurately compare hl-continuous and hl-discrete CFEs with low-level CFEs. Specifically, we align the individuals in both datasets to ensure a precise match. For example, individual one in the individual↦low-level CFE dataset corresponds directly to individual one in the individual↦hl-continuous CFE dataset, and so forth.

**Change in actions**  The metric, change in actions denoted as $\delta_{actions}(\cdot, \cdot)$ (Equation 7) assesses the difference in number of actions taken when an individual takes two different CFEs

$$\delta_{actions}(P, Q) = |P_{\text{actions}}| - |Q_{\text{actions}}| \tag{7}$$

where $P, Q$ are the two CFEs being considered and $|P_{\text{actions}}|$ and $|Q_{\text{actions}}|$ respectively, are the number of actions taken with the execution of each CFE.

**Change in improvement**  To compute the change in improvement $\delta_{improvement}(\cdot, \cdot)$, we first compute improvement, a distance between the initial state and resultant state (final state after taking a CFE) for each CFE. The change in improvement $\delta_{improvement}(\cdot, \cdot)$ is meant to assess the difference in how far agents change (improve) when they take two different CFEs, a low-level CFE and another CFE: hl-continuous or hl-discrete CFE.

$$P_{\text{improvement}} = \|\mathbf{x}' - \mathbf{x}\| \tag{8}$$

where $P$ is the CFE taken and $\mathbf{x}'$ is the resultant individual state after taking the CFE from $\mathbf{x}$, which is the initial individual state. Ideally high improvement (low proximity), $\mathbf{x}'$ more distant from $\mathbf{x}$ is preferred.

$$\delta_{improvement}(P, Q) = P_{\text{improvement}} - Q_{\text{improvement}} \tag{9}$$

Where $P, Q$ are the two CFEs and $P_{\text{improvement}}$ and $Q_{\text{improvement}}$ respectively, is the improvement achieved for taking the CFEs.

**Change in features**  We also compute the change in features $\delta_{features}(\cdot, \cdot)$ (Equation 10) to assess the difference in number of modified features when taking two different CFEs, a low-level CFE and another CFE: hl-continuous or hl-discrete CFE.

$$\delta_{features}(P, Q) = |P_{\text{features}}| - |Q_{\text{features}}| \tag{10}$$

Where $P, Q$ are the two CFEs and $|P_{\text{features}}|$ and $|Q_{\text{features}}|$ respectively, are the number of modified features with taking each CFE.

**Statistical significance between variables**  Given the different variables, e.g., list of the number of actions taken, number of modified features, and improvement achieved with each CFE: hl-continuous, hl-discrete CFEs, and low-level CFEs, we compute the statistical significance of the differences. We use the Scipy stats tool (Developers, 2023) to compute the Kendall tau and p-value to assess the statistical significance of difference, and the relationship between the two variables at a time.

**Coefficient of variation**  To assess how much the variables like number of modified features vary across groups, for example, between male and female individuals, we compute the coefficient of variations (Equation 11), a normalized measure of dispersion calculated as the ratio of the standard deviation to the mean.

$$\text{coefficient of variation}(V) = \frac{\text{standard deviation}_V}{\text{mean}_V} \times 100 \tag{11}$$

Where $V$ is the variable, such as number of actions taken by male and female negatively classified individuals.

## B DATASETS: SUPPLEMENTAL DETAILS

This section describes the supplemental details about the datasets used in the experiments.

### B.1 REAL-WORLD AND SEMI-SYNTHETIC DATASETS

First, we describe the extraction and preprocessing of real-world datasets: Foods, BMI, WHR, and BRFSS. Then, we describe the creation of semi-synthetic individual↦hl-continuous CFE, individual↦hl-discrete CFE, and individual↦hl-id CFE datasets.

#### B.1.1 FOODS, BMI, AND WHR DATASETS PREPROCESSING

**Intersectional nutritional features** After extracting the datasets for Foods, BMI, and WHR and removing features with missing values in the Foods dataset, we selected an intersectional subset of nutritional value features in the Foods and BMI datasets and the Foods and WHR datasets. This subset consisted of 20 features, including: *'protein (gm)', 'carbohydrate (gm)', 'dietary fiber (gm)', 'calcium (mg)', 'iron (mg)', 'magnesium (mg)', 'phosphorus (mg)', 'potassium (mg)', 'sodium (mg)', 'zinc (mg)', 'copper (mg)', 'selenium (mcg)', 'vitamin C (mg)', 'niacin (mg)', 'vitamin B6 (mg)', 'total folate (mcg)', 'vitamin B12 (mcg)', 'total saturated fatty acids (gm)', 'total monoun-saturated fatty acids (gm)'*, and *'total polyunsaturated fatty acids (gm)'*.

**Foods dataset preprocessing** The Foods dataset from Awram (2024) initially contained 53 features. After finding the intersectional subset of nutritional value features and removing datapoints with missing values, the dataset had 27 features. These included the following: *'NDB_No', 'Shrt_Desc', 'GmWt_1', 'GmWt_Desc1', 'GmWt_2', 'GmWt_Desc2'*, and *'Refuse_Pct'*, along with the 20 nutritional features described above. To add costs to the dataset, we web-scraped the average USD prices and extracted caloric prices for each food item given their name specified in the *'Shrt_Desc'* feature. Out of 3901 food items, we successfully extracted USD prices for 3871 food items and caloric prices for 3125 food items. Therefore, when using USD prices as costs, there were 3871 possible actions, while using caloric prices meant 3125 possible actions.

**BMI dataset preprocessing** The body mass index (BMI) dataset originally had 57 features. After removal of features with at least 20% null values and selecting the above nutritional features, except the feature *'total folate (mcg)'*, we had 23 features including: *'gender', 'age', 'race'*, and *'body mass index (kg/m**2)'*. We selected individuals whose age was greater than or equal to 20 at the time of surveys. Using the features *'body mass index (kg/m**2)'* and *'age'*, we computed the class for each individual as either healthy (1) BMI or unhealthy (0) (WebMD, 2024). We then removed the feature *'body mass index (kg/m**2)'* and all the duplicates datapoints. At the end of data preprocessing, we did the 80/20 train/test data split resulting in 40734 data points in the predictive training set and 10184 in the predictive testing set.

**WHR dataset preprocessing** Unlike the BMI dataset, there were fewer datapoints with 'waist-to-hip ratio' (WHR) information among the NHANES body measurement surveys (for years 1999 to prepandemic 2020) we scraped. First, we removed all features with at least 20% null values. Then using the features *'waist circumference (cm)', 'hip circumference (cm)'* and *'gender'*, we created the binary class variable *whr-class* (Wikipedia contributors, 2024), indicating healthy (1) or unhealthy (0) WHR. After preprocessing, we had 23 features, including the 20 nutritional features described above and the demographic features: *'gender', 'age'*, and *'race'*. Lastly, we removed the duplicates and split the dataset 80/20, creating 7296 data points in the predictive training set and 1824 in the predictive testing set.

#### B.1.2 BEHAVIORAL RISK FACTOR SURVEILLANCE SYSTEM (BRFSS) DATASET PREPROCESSING

The initial BRFSS dataset comprised 253680 rows and 22 features, each detailing various health and demographic attributes of individuals (Teboul, 2024b;a).

First, we removed all data points where '$Age$' $= 1$ denoting an age range of 18-24 because computation a new variable which relied on age being equal to or above 20 years, which reduced the dataset

to 247, 980 rows. The new variable was called '*HealthBMI*,' an adult health BMI classification value (WebMD, 2024) from the feature '*BMI*.' Next, we transformed the existing features, which were predominantly binary, into new features where the 1 represents a desirable condition and 0 otherwise. We focused particularly on features we deemed actionable and renamed them to enhance their intuitiveness, specific to satisfiability. For instance, we renamed the feature '*HighBP*', which indicated high blood pressure (0 = no, 1 = yes), to '*LowBP*': {1 = yes (lowBP), 0 = no (highBP)}. Additionally, we removed six features '*CholCheck*,' *Diabetes_012*,' '*Sex*,' '*Age*,' '*Education*,' and '*Income*,' and remained with 16 features.

These final 16 binary features included the following: '*LowBP*': {1 = yes (lowBP), 0 = no (highBP)}, '*LowChol*': {1 = yes (lowChol), 0 = no (highChol)}. The feature '*HealthBMI*': {1 =yes (healthy), 0 = no (unhealthy), '*NoSmoke*': {1 = yes, 0 = no}, '*NoStroke*': {1 = yes, 0 = no}, '*NoCHD*': {1 = yes, 0 = no}, '*PhysActivity*': {1 = yes, 0 = no}, '*Fruits*': {1 = yes, 0 = no}, '*Veggies*': {1 = yes, 0 = no}, '*LightAlcoholConsump*': {1 = yes, 0 = no}, '*AnyHealthcare*': {1 = yes, 0 = no}, '*DocbcCost*': {1 = yes, 0 = no}, '*GoodGenHlth*': {1 = excellent (1,2,3), 0 = bad (4,5)}, '*GoodMentHlth*': {1 = {1 = good ($< 2$), 0 = bad ($\geq 2$)}, '*GoodPhysHlth*': {1 = good ($< 2$), 0 = bad ($\geq 2$)}, and '*NoDiffWalk*': {1 = yes, 0 = no}.

Since we consider the setting where $\mathtt{t} = \mathbf{1}_{16}$, of the remaining data points, 8392 were considered to have a desirable outcome (no health risk) because all their features met the respective feature thresholds. Lastly, after removing the duplicate health risk individuals and splitting the whole dataset 80/20, we had 11039 data points in the predictive training set and 2760 in the predictive testing set.

### B.1.3 GENERATION OF THE SEMI-SYNTHETIC DATASETS

Below, we describe the creation of the four semi-synthetic, individual↦hl-continuous CFE datasets: BMI and WHR individual states with either monetary or caloric cost actions. Additionally, we provide details of generating the one individual↦hl-discrete CFE dataset: BRFSS with synthetic hl-discrete actions. Finally, we detail the creation of the derivative individual↦hl-id CFE datasets. Figure 3 shows examples of generated hl-continuous CFEs for BMI and WHR individual states, and an hl-discrete CFE for a BRFSS individual state.

After creating the individual↦hl-continuous CFE, individual↦hl-discrete CFE, and the individual↦hl-id CFE datasets, we trained and tested the corresponding CFE generators. For instance, we trained and tested the data-driven hl-continuous CFE generator using the individual↦hl-continuous CFE datasets. We conducted all experiments on a laptop with a CPU featuring the following hardware specifications: a 2.6 GHz 6-Core Intel Core i7 processor, 16 GB of 2400 MHz DDR4 RAM, and an Intel UHD Graphics 630 with 1536 MB of video memory.

**The individual↦hl-continuous CFE datasets** Using the BMI, WHR, and Foods+Costs (monetary and caloric) datasets, we generated four distinct individual↦hl-continuous CFE datasets. First, we trained classification models to identify individuals who required CFEs. For both the BMI and WHR datasets, we hyperparameter-tuned the *solver* and *max_iter* parameters of logistic regression models using their respective training predictive data. The respective best logistic regression models achieved a test accuracy of 72.78% on the BMI dataset, 85.18% on the WHR dataset and 100.00% on BRFSS dataset. Based on these models, we determined the model prediction outcome for all individuals in the training and testing sets.

After identifying negatively classified individuals in the training and test sets, we computed their respective hl-continuous CFEs. We considered two types of actions: Foods with either monetary costs or caloric costs. For the negatively classified individuals and given the classifier model parameters (coefficients and intercepts) and hl-continuous actions, we used the ILP (see Equation 2) to generate two types of hl-continuous CFEs for each individual: one optimized for caloric cost and the other for monetary cost actions.

Consequently, we generated four distinct individual↦hl-continuous CFE datasets. Each dataset comprises hl-continuous CFEs characterized by Foods and their associated costs, which can be either monetary or caloric, optimized accordingly. For the BMI dataset, we generated two individual↦hl-continuous CFE datasets, 40692 for the training set and 10167 for the test set. With similar statistics, in one, the hl-continuous CFE result of optimization with the food+monetary cost actions, and another from the food+caloric cost actions. Likewise, for the WHR dataset, we generated 6387 training

set, and 1603 test set individual↦hl-continuous CFEs datasets with actions described by Foods and monetary costs, and the same with actions defined by Foods and caloric costs.

|  | action-1 | action-2 |
|---|---|---|
| **PhysActivity** | 0 | 1 |
| **Fruits** | 0 | 1 |
| **Veggies** | 0 | 1 |
| **AnyHealthcare** | 0 | 0 |
| **LowBP** | 1 | 1 |
| **NoSmoke** | 0 | 1 |
| **LowChol** | 1 | 0 |
| **HealthBMI** | 0 | 1 |
| **NoStroke** | 1 | 0 |
| **NoCHD** | 0 | 0 |
| **LightAlcoholConsump** | 0 | 1 |
| **DocbcCost** | 1 | 0 |
| **GoodGenHlth** | 0 | 0 |
| **GoodMentHlth** | 1 | 0 |
| **GoodPhysHlth** | 1 | 0 |
| **NoDiffWalk** | 1 | 0 |

(a) for a BRFSS individual state

|  | action-1 | action-2 | action-3 |
|---|---|---|---|
| Protein (gm) | 1.800 | 0.000 | 0.400 |
| Carbohydrate (gm) | 3.740 | 61.500 | 0.100 |
| Dietary fiber (gm) | 1.600 | 0.200 | 0.000 |
| Calcium (mg) | 51.000 | 8.000 | 13.000 |
| Iron (mg) | 1.800 | 3.720 | 0.300 |
| Magnesium (mg) | 81.000 | 2.000 | 11.000 |
| Phosphorus (mg) | 46.000 | 5.000 | 114.000 |
| Potassium (mg) | 379.000 | 16500.000 | 149.000 |
| Sodium (mg) | 213.000 | 52.000 | 215.000 |
| Zinc (mg) | 0.360 | 0.420 | 0.100 |
| Copper (mg) | 0.179 | 0.195 | 0.389 |
| Selenium (mcg) | 0.900 | 0.200 | 4.100 |
| Vitamin C (mg) | 30.000 | 0.000 | 1.000 |
| Niacin (mg) | 0.400 | 0.000 | 0.180 |
| Vitamin B6 (mg) | 0.099 | 0.000 | 0.010 |
| Vitamin B12 (mcg) | 0.000 | 0.000 | 5.000 |
| Total saturated fatty acids (gm) | 0.030 | 0.000 | 0.002 |
| Total monounsaturated fatty acids (gm) | 0.040 | 0.000 | 0.002 |
| Total polyunsaturated fatty acids (gm) | 0.070 | 0.000 | 0.006 |

(b) for a BMI individual state

|  | action-1 | action-2 |
|---|---|---|
| Protein (gm) | 0.000 | 19.440 |
| Carbohydrate (gm) | 61.500 | 0.000 |
| Dietary fiber (gm) | 0.200 | 0.000 |
| Calcium (mg) | 8.000 | 17.000 |
| Iron (mg) | 3.720 | 1.630 |
| Magnesium (mg) | 2.000 | 23.000 |
| Phosphorus (mg) | 5.000 | 139.000 |
| Potassium (mg) | 16500.000 | 179.000 |
| Sodium (mg) | 52.000 | 247.000 |
| Zinc (mg) | 0.420 | 0.690 |
| Copper (mg) | 0.195 | 0.050 |
| Selenium (mcg) | 0.200 | 70.600 |
| Vitamin C (mg) | 0.000 | 0.000 |
| Niacin (mg) | 0.000 | 10.136 |
| Vitamin B6 (mg) | 0.000 | 0.319 |
| Total folate (mcg) | 0.000 | 4.000 |
| Vitamin B12 (mcg) | 0.000 | 2.550 |
| Total saturated fatty acids (gm) | 0.000 | 0.211 |
| Total monounsaturated fatty acids (gm) | 0.000 | 0.107 |
| Total polyunsaturated fatty acids (gm) | 0.000 | 0.277 |

(c) for a WHR individual state

Figure 3: In (a), for an individual negatively classified based on their BRFSS features, with values $[0, 0, 0, 1, 0, 0, 0, 0, 0, 1, 1, 1, 1, 1, 0, 0]$ arranged similarly to the features in (a), the hl-discrete CFE generator recommends hl-discrete actions, specifically **action-1** and **action-2**.
In (b), for a negatively classified BMI individual, given their actionable features with values $[253.51, 352.76, 48.2, 1327., 29.61, 1204., 3966., 6163., 5890.0, 44.19, 7.903, 275.1, 30., 109.198, 3.492, 2.3, 59.686, 154.24, 113.429]$, arranged in the same order as features shown in (b), the hl-continuous CFE generator recommends a CFE containing the following hl-continuous actions: **action-1** (*take Swiss chard, raw*), **action-2** (*take leavening agents: cream of tartar*), and **action-3** (*take clams, mixed species, canned, in liquid*).
Similarly, in (c), for a negatively classified WHR individual with actionable feature values $[29.03, 109.45, 4.1, 309., 4.08, 96., 488., 994., 1326., 2.61, 0.425, 45., 35.7, 8.755, 0.482, 172., 1.21, 10.077, 12.392, 13.999]$, ordered as features in (c), the hl-continuous CFE generator recommends a CFE with the following hl-continuous actions: **action-1** (*take leavening agents: cream of tartar*) and **action-2** (*take fish, tuna, light, canned in water, drained solids*).

**The individual↦hl-discrete CFE datasets** First, we generated 100 synthetic actions, each of length 16. We set the probability $p_a$ of an action fulfilling the capability of a given at 0.5. The costs associated with fulfilling the capabilities of each feature were randomly predefined and were uniform across all actions and individuals. We computed the cost of an action as the sum of the costs of adding capabilities to individual features.

Given the BRFSS dataset, synthetic actions, and the unit threshold-based binary classifier $\mathbf{t} = \mathbf{1}_n$, we used the ILP (see Equation 3) to generate an individual↦hl-discrete CFE dataset. At the end, we had 11039 train-set and 2760 test set BRFSS with synthetic actions individual↦hl-discrete CFE dataset.

**The individual↦hl-id CFE datasets** Given the individual↦hl-continuous CFE and individual↦hl-discrete CFE datasets described earlier, we created corresponding individual↦hl-id CFE datasets. This process involves encoding each CFE in the individual↦CFE dataset with a unique identifier that distinguishes it from all other possible CFEs in that dataset. For example, given instances of individuals–hl-discrete CFEs, we generate unique identifiers for all the hl-discrete CFEs to generate corresponding hl-id CFEs.

**The semi-synthetic varied frequency of CFEs datasets** Before the train/test individual↦CFE datasets split, for each of the generated individual↦hl-continuous CFE, individual↦hl-discrete CFE, and the individual↦hl-id CFE datasets, we generate three frequency of CFE dataset variants: `all` (including all data), `>10` (more than 10 individuals per CFE), and `>40` (more than 40 individuals per CFE).

## B.2 FULLY-SYNTHETIC DATASETS

We created five variants of the synthetic individual↦hl-discrete CFE datasets: varied dimension, frequency of CFEs, information access, feature satisfiability, and actions access. We provide statistical detailed information about the five variations of the individual↦hl-discrete CFE datasets in Table 2 and Figure 4.

### B.2.1 VARIED DIMENSIONS

We created 20-, 50- and 100-dimensional individual datasets by varying the number of actionable features ($n = 20, 50, 100$) and keeping $p_f = 0.68$ the same for all datasets. We consider a unit vector threshold of length $n$. The cost associated with fulfilling the capabilities of each feature was predefined randomly and the same across all actions and individuals. Each action was of length $n$, $p_a$ was 0.5, and action cost was the sum of the cost for each features the action fulfills. To create the 20-, 50- and 100-dimensional individual↦hl-discrete CFE datasets, we computed the hl-discrete CFEs for each varied dimensional dataset individual using the information above and the ILP defined in Equation 3 using `CVXPY` Python package (Diamond & Boyd, 2016; Agrawal et al., 2018).

### B.2.2 VARIED FREQUENCY OF CFES

To investigate the effect of frequency of CFEs in the individual↦CFE training set on the performance of the data-driven CFE generator, we create the varied frequency of CFEs variant datasets. For each of the varied dimensional individual↦hl-discrete CFE datasets described in Appendix B.2.1, before the train/test split, we created three frequency-based dataset variants: `all`, where all data is included, `>10`, where we ensure a frequency of more than 10 individuals per hl-discrete CFE, and `>40` with insurance of a frequency of more than 40 individuals per hl-discrete CFE.

### B.2.3 VARIED INFORMATION ACCESS

In addition to general "*varied information access*" variants that we considered: individual↦hl-continuous CFE, individual↦hl-discrete CFE, and the individual↦hl-id CFE datasets, we investigate more settings derived from the fully synthetic hl-discrete CFEs.

For each of the 20-, 50- and 100-dimensional individual↦hl-discrete CFE datasets and their corresponding frequency-based datasets (`all`, `>10`, and `>40`), we created three "*varied information*

| Dataset name | Dataset size | One-action CFEs | Two-action CFEs | Three-action CFEs |
|---|---|---|---|---|
| 20-dimensional dataset | 71125 | 23687 | 44858 | 2576 |
| 50-dimensional dataset | 98966 | 1262 | 96770 | 934 |
| 100-dimensional dataset | 99728 | 0 | 45515 | 54213 |
| Manual groups | 73484 | 13480 | 56653 | 3351 |
| Probabilistic groups | 70226 | 44661 | 20258 | 5307 |
| First10 | 74524 | 61794 | 12046 | 39 |
| First5 | 74594 | 60656 | 6005 | 0 |
| Last10 | 74401 | 53822 | 19952 | 1 |
| Last5 | 74565 | 66068 | 644 | 0 |
| Mid5 | 74594 | 63530 | 3010 | 0 |

Table 2: Statistics of the individual↦hl-discrete CFE variant datasets used in the experiments. Each individual in all datasets has atmost 3 hl-discrete actions in their CFE.

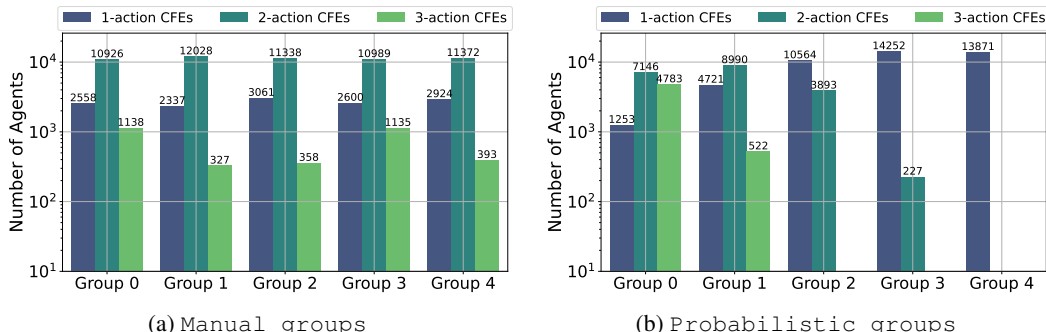

(a) Manual groups

(b) Probabilistic groups

Figure 4: Statistics of the "*varied actions access*" datasets for Manual groups and Probabilistic groups. For the Probabilistic groups, Group 0 is $p_a = 0.4$, Group 1 is $p_a = 0.5$, Group 2 is $p_a = 0.6$, Group 3 is $p_a = 0.7$, and Group 4 is $p_a = 0.8$.

*access*" datasets variants to represent the hl-discrete CFEs: the original hl-discrete CFE, left unchanged; the *hl-discrete-named* CFE, where a unique name encodes each hl-discrete action in the hl-discrete CFE; and the *hl-discrete-id* CFE, where a unique identifier denotes the entire hl-discrete CFE. For example, consider an individual $\mathbf{x} = [0, 0, 0, 0, 1]$ and their corresponding hl-discrete CFE given by $\{[0, 0, 1, 1, 0], [0, 1, 0, 0, 0], [1, 0, 0, 0, 0]\}$. The hl-discrete-named CFE is then given by $\{a, b, c\}$ where each hl-discrete action has a name/label (e.g., $a$) that uniquely identifies a specific hl-discrete action (e.g., $[0, 0, 1, 1, 0]$) among all hl-discrete actions. On the other hand, a unique name, say $z$, denotes the hl-discrete-id CFE, where $z$ uniquely represents this specific hl-discrete CFE among all the hl-discrete CFEs.

This setting aims to study the effectiveness of the data-driven CFE generators under various information access constraints within an individual↦CFE training set, for example, (1) full access to hl-discrete actions and their effects on features (hl-discrete CFE), (2) access only to the names of hl-discrete actions without any information on how each action affects features (hl-discrete-named CFE), and (3) minimal information access, where only hl-discrete-id CFEs are known, with no explicit knowledge of the corresponding hl-discrete actions or their impact on features.

Given the individual↦hl-discrete CFE "*varied information access*" datasets, we use the data-driven CFE generator architectures described in Section 4.2 to generate the CFEs. Specifically, we use the hl-discrete CFE generator to generate hl-discrete CFEs, hl-continuous CFE generators for hl-discrete-named CFEs, and hl-id CFE generators for hl-discrete-id CFEs.

### B.2.4   VARIED FEATURE SATISFIABILITY

Using the ILP formulation defined in Equation 3 with $n = 20$, and following the same individual and hl-discrete generation approach as in Appendix B.2.1 while varying the feature satisfiability for the threshold-based binary classifier (differing in which features are classifier-active (non-zero)), we generated five individual$\mapsto$hl-discrete CFE datasets. For the dataset Last5, the threshold vector is set as $\mathbf{t} = [15 \text{ zeros}, 5 \text{ ones}]$, while for the dataset First5, it is set as $\mathbf{t} = [5 \text{ ones}, 15 \text{ zeros}]$. The third dataset, First10, has a threshold vector of $\mathbf{t} = [10 \text{ ones}, 5 \text{ zeros}]$, and the dataset Last10 has $\mathbf{t} = [10 \text{ zeros}, 10 \text{ ones}]$. Finally, the dataset Mid5 has all features set to zero except for the five middle features set to one.

These "*varied feature satisfiability*" variants of the individual$\mapsto$hl-discrete CFE datasets are specifically created to investigate the effect of feature satisfiability on the nature of the hl-discrete CFEs and the effectiveness of the hl-discrete CFE generator at generating CFEs for new individuals.

### B.2.5   VARIED ACCESS TO ACTIONS

Lastly, we consider two settings where grouped individuals have restricted access to a set of actions: 1) manual groups where actions generated with the same probability $p_a = 0.5$ and individuals are randomly assigned a restricted subset of actions; and 2) probabilistic groups where individuals are assigned to groups and each group has its actions generated by different probabilities $p_a = [0.4, 0.5, 0.6, 0.7, 0.8]$. See Figure 4 for the statistics of the datasets.

We designed the "varied access to actions" variants to empirically investigate fairness in CFE generation. Specifically, we examine the impact of restricting access of a group of individuals to some actions on the characteristics of hl-discrete CFEs, such as their associated costs and the variations in accuracy of hl-discrete CFE generators across different groups.

## C CFE GENERATION: SUPPLEMENTAL DETAILS

Below we provide the supplemental detailed information on the experimental setups and methodology for generation of CFEs, using the proposed data-driven CFE generators and the low-level CFE generator (actionable recourse) (Ustun et al., 2019).

### C.1 THE LOW-LEVEL CFE GENERATOR

To compare the low-level CFE generators with the proposed data-driven CFE generators, we first generate low-level CFEs (see examples in Figure 5) for individuals who were negatively classified in the BMI, WHR, and BRFSS datasets, using Equation 1.

For all datasets, to determine which individuals require CFEs, we use the classification models detailed in Appendix B.1.3. Additionally, we employ the same actionable features as those used for generating hl-continuous CFEs for the BMI and WHR negatively classified individuals and hl-discrete CFEs generation for the BRFSS negatively classified individuals.

**BMI actionable features**   For BMI individuals states, we considered the following **19** actionable features: '*protein (gm)*', '*carbohydrate (gm)*', '*dietary fiber (gm)*', '*calcium (mg)*', '*iron (mg)*', '*magnesium (mg)*', '*phosphorus (mg)*', '*potassium (mg)*', '*sodium (mg)*', '*zinc (mg)*', '*copper (mg)*', '*selenium (mcg)*', '*vitamin C (mg)*', '*niacin (mg)*', '*vitamin B6 (mg)*', '*vitamin B12 (mcg)*', '*total saturated fatty acids (gm)*', '*total monounsaturated fatty acids (gm)*', and '*total polyunsaturated fatty acids (gm)*'.

**WHR actionable features**   For the generation of recourse for WHR individuals, we use the following **20** actionable features: '*protein (gm)*', '*carbohydrate (gm)*', '*dietary fiber (gm)*', '*calcium (mg)*', '*iron (mg)*', '*magnesium (mg)*', '*phosphorus (mg)*', '*potassium (mg)*', '*sodium (mg)*', '*zinc (mg)*', '*copper (mg)*', '*selenium (mcg)*', '*vitamin C (mg)*', '*niacin (mg)*', '*vitamin B6 (mg)*', '*total folate (mcg)*', '*vitamin B12 (mcg)*', '*total saturated fatty acids (gm)*', '*total monounsaturated fatty acids (gm)*', and '*total polyunsaturated fatty acids (gm)*'.

**BRFSS actionable features**   Lastly, for the BRFSS individual states, we considered the following **16** actionable features: '*PhysActivity*', '*Fruits*', '*Veggies*', '*AnyHealthcare*', '*LowBP*', '*NoSmoke*', '*LowChol*', '*HealthBMI*', '*NoStroke*', '*NoCHD*', '*LightAlcoholConsump*', '*DocbcCost*', '*GoodGenHlth*', '*GoodMentHlth*', '*GoodPhysHlth*', and '*NoDiffWalk*'.

Refer to Appendix B.1.1 and Appendix B.1.2 for a detailed dscription of the meaning of the features.

### C.2 DATA-DRIVEN CFE GENERATORS ARCHITECTURES: SUPPLEMENTAL DETAILS

This section includes supplemental details about the architectures of the data-driven CFE generators and information about other baseline models.

#### C.2.1 THE HL-CONTINUOUS CFE GENERATOR

The neural-network hl-continuous CFE generator we use in these experiments is susceptible to imbalance and overfitting. Therefore, we weight and regularize the loss function $\mathcal{L}_{\text{FA}}$ in Equation 5 as follows:

$$\mathcal{L}_{\text{FA}}^{w} = p_w \mathcal{L}_{\text{FA}} + \alpha \frac{1}{M} \sum_{m=1}^{M} ||\hat{a}_m - a_m||_1 \tag{12}$$

The weighting factor $p_w$ weights $\mathcal{L}_{\text{FA}}$ by scaling the contribution of each individual to the loss function. The term $\alpha \frac{1}{M} \sum_{m=1}^{M} ||\hat{a}_m - a_m||_1$ regularizes the model, thus preventing overfitting by nudging the model towards producing hl-continuous CFEs closer to $a_m$'s distribution. We, on average chose the values of $\alpha$ from the set $\{0.05, 0.1, 0.07\}$ and $p_w$ from $\{0.05, 0.1, 0.07\}$.

| Features to Change | Current Value | to | Required Value |
|---|---|---|---|
| Protein (gm) | 253.51 | → | 14.639999999999986 |
| Calcium (mg) | 1327 | → | 116 |
| Iron (mg) | 29.61 | → | 34.842000000000006 |
| Potassium (mg) | 6163 | → | 6370.618584999997 |
| Selenium (mcg) | 275.1 | → | 313.9095759999997 |
| Total monounsaturated fatty acids (gm) | 154.24 | → | 88.88112600000001 |

(a) for a BMI individual state

| Features to Change | Current Value | to | Required Value |
|---|---|---|---|
| Selenium (mcg) | 45 | → | 327.7319 |
| Total monounsaturated fatty acids (gm) | 12.392 | → | 89.34236700000017 |
| Total saturated fatty acids (gm) | 10.077 | → | 2.6004500000000004 |
| Vitamin B12 (mcg) | 1.21 | → | 0.1200000000000001 |
| Total folate (mcg) | 172 | → | 1179.7380000000003 |
| Vitamin B6 (mg) | 0.482 | → | 0.21794999999999998 |
| Niacin (mg) | 8.755 | → | 85.10721500000001 |
| Vitamin C (mg) | 35.7 | → | 0.10000000000000142 |
| Copper (mg) | 0.425 | → | 0.212950000000001 |
| Zinc (mg) | 2.61 | → | 1.3895 |
| Sodium (mg) | 1326 | → | 626.65 |
| Potassium (mg) | 994 | → | 6520.550000000004 |
| Phosphorus (mg) | 488 | → | 217 |
| Magnesium (mg) | 96 | → | 57 |
| Iron (mg) | 4.08 | → | 42.72946000000002 |
| Calcium (mg) | 309 | → | 113 |
| Total polyunsaturated fatty acids (gm) | 13.999 | → | 4.40896 |
| Carbohydrate (gm) | 109.45 | → | 43.376000000000005 |
| Dietary fiber (gm) | 4.1 | → | 50.113950000000024 |

| Features to Change | Current Value | to | Required Value |
|---|---|---|---|
| PhysActivity | 0 | → | 1 |
| Fruits | 0 | → | 1 |
| Veggies | 0 | → | 1 |
| LowBP | 0 | → | 1 |
| NoSmoke | 0 | → | 1 |
| LowChol | 0 | → | 1 |
| HealthBMI | 0 | → | 1 |
| NoStroke | 0 | → | 1 |
| GoodPhysHlth | 0 | → | 1 |
| NoDiffWalk | 0 | → | 1 |

(b) for a WHR individual state                (c) for a BRFSS individual state

Figure 5: In (a), for a negatively classified a BMI individual, given their actionable features, with values [253.51, 352.76, 48.2, 1327., 29.61, 1204., 3966., 6163., 5890.0, 44.19, 7.903, 275.1, 30., 109.198, 3.492, 2.3, 59.686, 154.24, 113.429] presented in the order specified in Appendix C.1, the low-level CFE generator recommends the CFE shown in (a). In (b), given actionable features values [29.03, 109.45, 4.1, 309., 4.08, 96., 488., 994., 1326., 2.61, 0.425, 45., 35.7, 8.755, 0.482, 172., 1.21, 10.077, 12.392, 13.999] in the order as described in Appendix C.1 for a negatively classified WHR individual, the low-level CFE generator recommends the CFE shown in (b). Finally, in (c), for an individual negatively classified based on their BRFSS features, with values [0, 0, 0, 1, 0, 0, 0, 0, 0, 1, 1, 1, 1, 1, 0, 0] in the order described below, the low-level CFE generator recommends the CFE shown in (c).

### C.2.2 THE HAMMING DISTANCE CFE GENERATOR

To produce hl-discrete-id CFEs (refer to Appendix B.2.3) for new individuals, we mainly used the hl-id CFE generator. However, we wanted to investigate the effect of model complexity on the accuracy of CFE generation. Therefore, we compare the more complex hl-id CFE generator (refer to Section 4.2) with a basic model, e.g., Hamming distance-based CFE generator, whose choice is due to the individual features being binary for this setting. Below is a description of the Hamming distance hl-discrete-id CFE generator.

Given a negatively classified new individual $\mathbf{x}_{ts}$, we compute the Hamming distance (see Figure 7) between them and each of the individuals $\mathbf{x}_{tr}$ in the individual↦hl-discrete-id CFE training set.

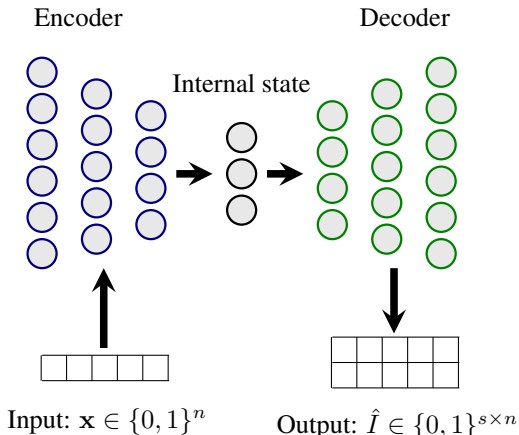

Figure 6: An encoder-decoder hl-discrete CFE generator, where $n$ is the dimension and $s$ is the number of hl-discrete actions in the CFE.

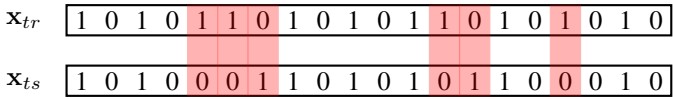

Hamming Distance: $(\mathbf{x}_{tr}, \mathbf{x}_{ts}) = 6$

Figure 7: Hamming distance between a training set individual $\mathbf{x}_{tr}$ and a new individual $\mathbf{x}_{ts}$.

Then, based on these distances, we choose the $k$ nearest training set individuals and their associated hl-discrete CFEs. We then use the most common hl-discrete-id CFE as the hl-discrete-id CFE for the new individual $\mathbf{x}_{ts}$. We experimented with varied number of nearest neighbours: $5, 10$ and $15$, for the 20-, 50- and 100-dimensional individual$\mapsto$hl-discrete-id CFE datasets, respectively.

# D  EXPERIMENTAL RESULTS: SUPPLEMENTAL DETAILS

In this section, we provide additional and thorough empirical evidence demonstrating the strong performance of the proposed data-driven CFE generators in producing optimal CFEs for new individuals. We also show how they address the challenges associated with low-level CFE generators. Specifically, we highlight the strong and desirable characteristics of the hl-continuous and hl-discrete CFEs in comparison to low-level CFEs. We also analyze how various constraints–such as varied data dimensions, the frequency of CFEs, decision-makers information access, feature satisfiability, and restrictions on individuals' access to actions—affect the individual↦CFE data distribution and the effectiveness of data-driven CFE generators.

## D.1  LEAD TO DIVERSE AND HIGHER IMPROVEMENT

Unlike low-level CFEs, using hl-continuous and hl-discrete CFEs generally requires fewer actions on average (see Figure 8(a)). These CFEs also lead to more diverse improvements, simultaneously modifying multiple features (see Figures 8(b) and 9(a)) and resulting in states that are significantly further from the initial state (Figures 8(c) and 9(b)).

**(a)**

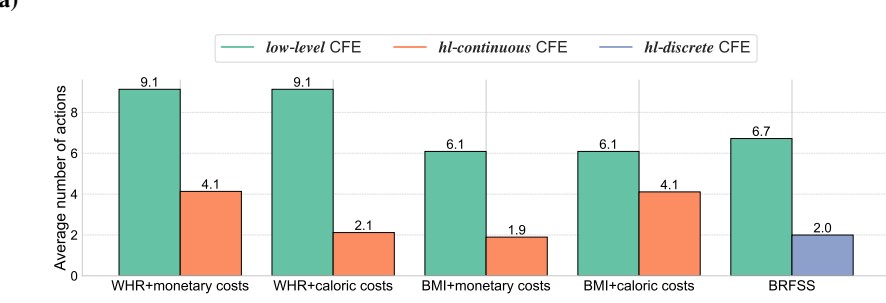

**(b)**

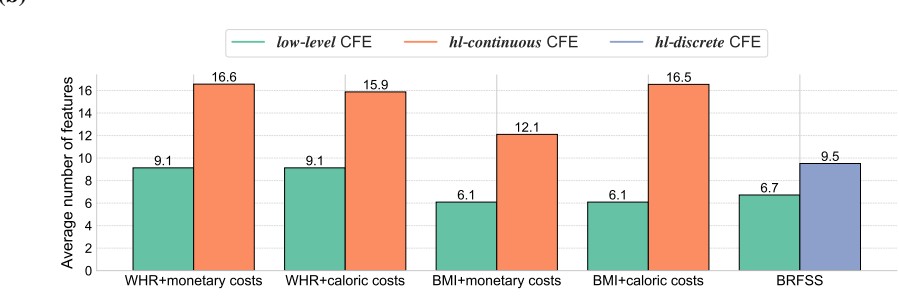

**(c)**

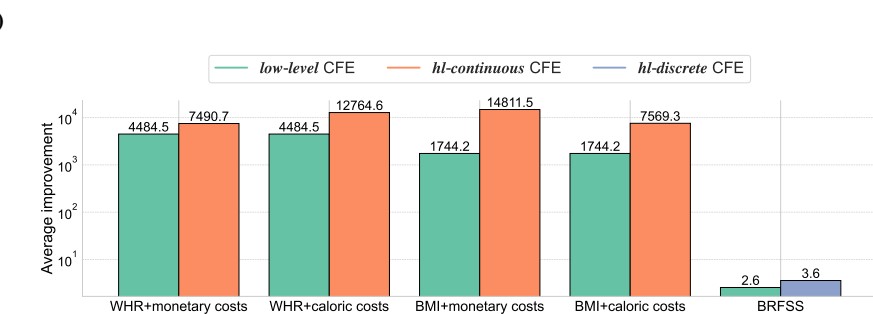

Figure 8: All figure annotations rounded to one decimal place, the figures show the comparison of hl-continuous CFEs (monetary and caloric costs) for WHR and BMI datasets and hl-discrete CFEs on BRFSS dataset with the low-level CFEs on respective datasets. Results show that taking low-level CFEs involves (a) more actions, (b) fewer feature modifications, and (c) less improvement (closer resultant (new) states), than hl-discrete and hl-continuous CFEs.

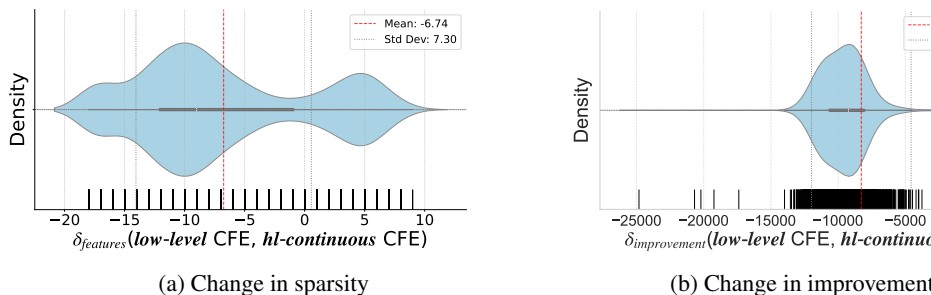

(a) Change in sparsity

(b) Change in improvement

Figure 9: Given WHR negatively classified individuals and the low-level and hl-continuous CFEs they took, a computation of $\delta_{improvement}(P,Q)$ (Equation 9)} and $\delta_{features}(P,Q)$ (Equation 10) where $P$ denotes taking a low-level CFE and $Q$ denotes taking an hl-continuous CFE, shows that when individuals take hl-continuous CFEs, a higher number of their features is modified (a) and their improvement is significantly higher (b) than if they took low-level CFEs.

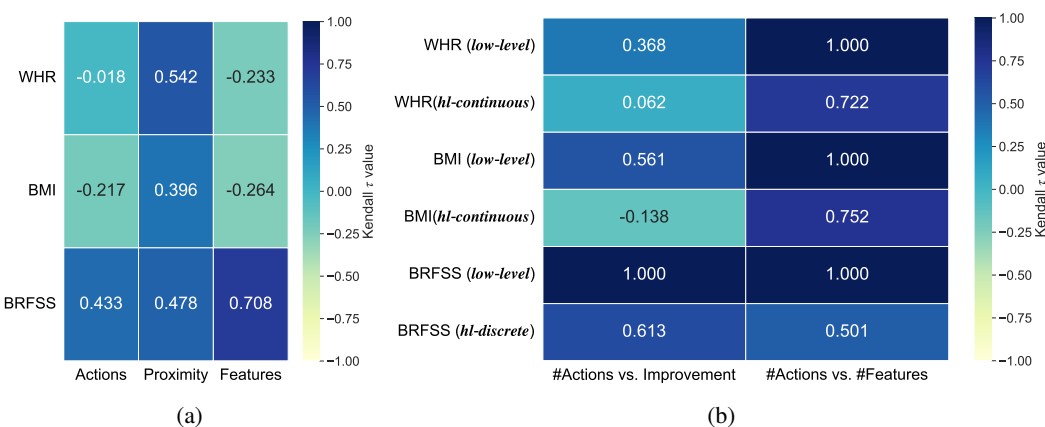

Figure 10: In (a), we illustrate the correlations for three different aspects: (1) between the number of actions taken with CFEs P and Q, (2) between the number of features modified with CFEs P and Q, and (3) between the improvement achieved after taking CFEs P and Q. For the BMI and WHR datasets, P and Q represent low-level and hl-continuous CFEs, respectively. For the BRFSS dataset, P and Q denote low-level and hl-discrete CFEs, respectively. On the other hand, (b) shows the correlation between the number of actions taken and the number of modified features and between the number of actions taken and improvement achieved for each CFE and dataset. In general, low-level CFEs have a perfect positive relationship between the number of actions and modified features.

Moreover, while low-level CFEs exhibit a perfect correlation between the number of actions taken and the number of features modified, as shown in Figure 10(b), hl-continuous and hl-discrete CFEs display a positive but weaker relationship. This imperfect correlation is often more desirable as it better reflects real-world scenarios, and ideally, one wants to make more changes with fewer and interpretable actions. Additionally, there was a high positive correlation ($\tau = 0.708$) between number of modified features with hl-discrete and low-level CFEs (see Figure 10(a)). In general, there was a weak negative correlation between number of modified feature with hl-continuous and low-level CFEs, and between number of actions taken with hl-continuous and low-level CFEs.

## D.2 EASIER TO PERSONALIZE AND INTERROGATE FAIRNESS

Fairness in CFE generation has primarily been studied along the dimension of equalizing the recourse costs across different groups (e.g, (Gupta et al., 2019b)). In this work, we extend the analysis by exploring several dimensions of fairness in CFE generation. First, we assess the variability outcome of CFEs execution. Specifically, we investigate how individuals using the same CFE generator

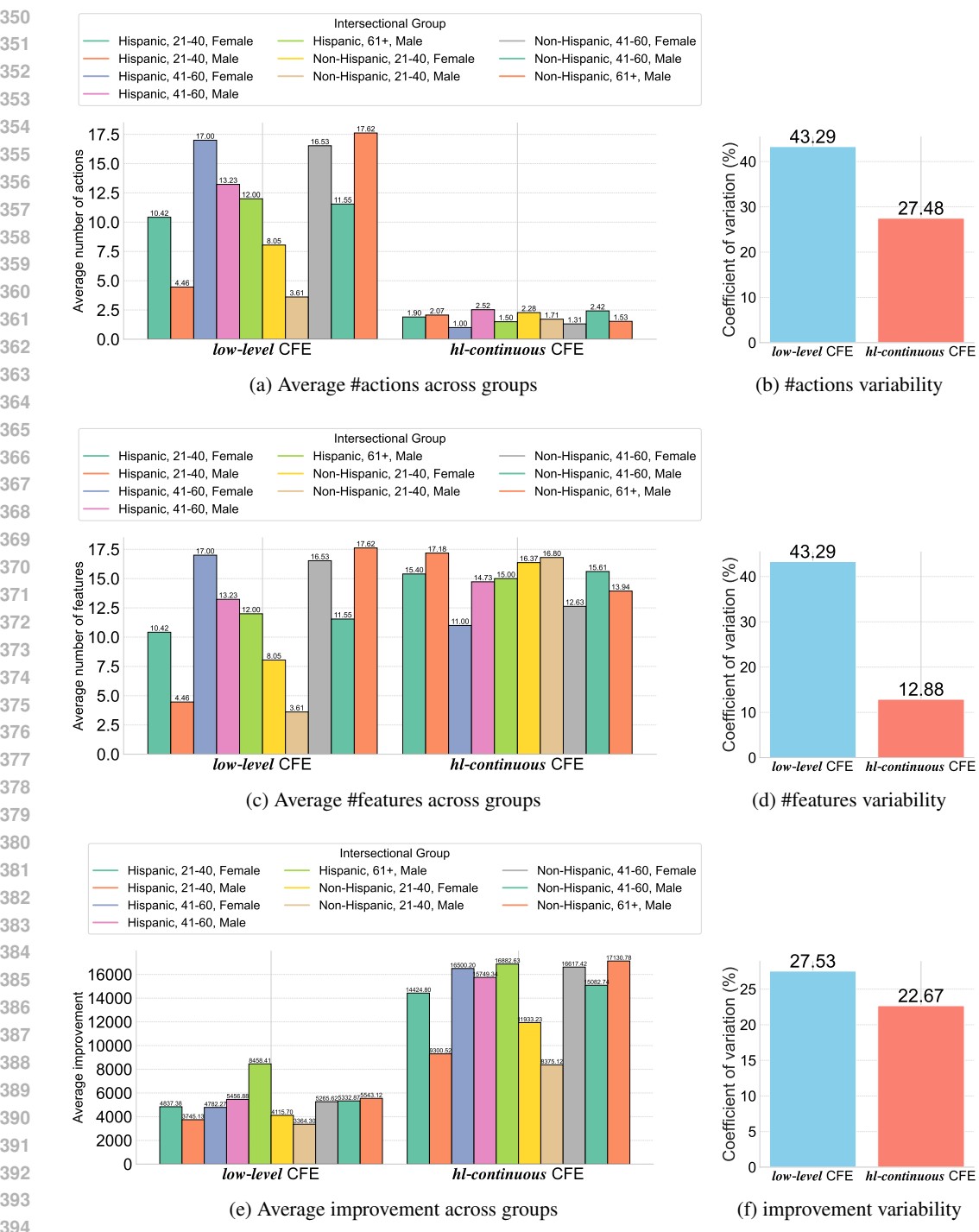

(a) Average #actions across groups

(b) #actions variability

(c) Average #features across groups

(d) #features variability

(e) Average improvement across groups

(f) improvement variability

Figure 11: The figures illustrate the variations in three variables: the average number of actions taken, the number of features modified, and the improvement achieved by individuals from different sensitive groups when using the same type of CFE, such as low-level or hl-continuous CFEs. (a), (c), and (e) depict the distributions for these variables across sensitive groups. To better assess variability, (b), (d), and (f) present the coefficients of variation that concisely illustrate the extent of dispersion around the mean. All figures indicate that low-level CFEs are less fair than hl-continuous CFEs, as the latter have lower coefficients of variation across all variables, which means that agents from different sensitive groups are more likely to achieve close to similar outcomes when they take hl-continuous CFEs.

(same kind of CFEs) experience differences in how much they improve, the number of actions taken, the number of modified features, and the costs incurred, particularly across sensitive groups. Second, we explore the effects of limiting access to a subset of actions ("*varied access to actions*") on the distribution of individual$\mapsto$CFE datasets and the accuracy of CFE generators across different groups. Lastly, we examine how classification models or predetermined actionable features ("*varied feature satisfiability*") influence the distribution of the individual$\mapsto$CFE dataset and the performance of generators on different groups.

In addition to fairness, we also investigate the personalization of CFE generation along two dimensions. 1) Individuals may be interested in a subset of actions ("*varied access to actions*") and thus restricted to CFEs that involve only specific actions. 2) Individuals might prioritize different costs in the generation process ("*varied cost preferences*") and thus prefer CFE generators that optimize those specific costs in CFE generation, e.g., caloric costs over monetary ones.

Below is the detailed empirical evidence on how hl-continuous and hl-discrete CFEs are easier to personalize and how their generators are easier to interrogate for fairness issues.

### D.2.1 FAIRNESS BASED ON VARIABILITY OF CFES EXECUTION OUTCOME

We investigate variation in costs incurred and individual improvement (number of actions taken, number of features modified, and improvement) across intersectional sensitive groups to understand how the fairness of the low-level CFE generators compares to that of hl-continuous CFE and hl-discrete CFE generators.

**Variability in individual improvement across sensitive groups**    We investigate variations in improvement by studying the differences in improvement, i.e., how far the resultant state is from the initial state (proximity), diversity of improvement, i.e., how many features the CFE modifies, and ease of improvement, i.e., number of actions taken, across sensitive groups.

Figure 11 shows that on the WHR dataset, using low-level CFEs led to significant variation in improvement across sensitive groups, specific to how much individuals improve, the number of actions taken, and the number of features modified. Specifically, variations with taking low-level CFEs versus low-level are such that the coefficient of variation for how much individual improve was $27.53\%$ compared to $22.67\%$, for average number of actions taken it was $43.29\%$ compared to $27.48\%$, and for modified features it was $43.29\%$ compared to $12.88\%$. These findings highlight that the benefits of low-level CFEs differ substantially across sensitive groups, potentially favoring some over others, a potential fairness issue in CFE generation.

**Variability in costs incurred across sensitive groups**    Although the costs individuals incur by taking low-level CFEs cannot be directly compared with taking hl-continuous CFEs because they are contextually different, we study how the costs of executing the same kind of CFEs varies across individuals in different sensitive groups.

Our results show that taking low-level CFEs varies more widely across various sensitive groups than taking hl-continuous CFEs. For example, in Figure 12, the coefficient of variation for taking low-level CFEs is $41.16\%$ and $79.55\%$ versus $5.60\%$ and $37.61\%$ with taking hl-continuous CFEs, on BMI and WHR datasets , respectively. Therefore, compared to taking hl-continuous CFEs, taking low-level CFEs is more biased and more likely to cost-wise favor some sensitive groups over others than taking.

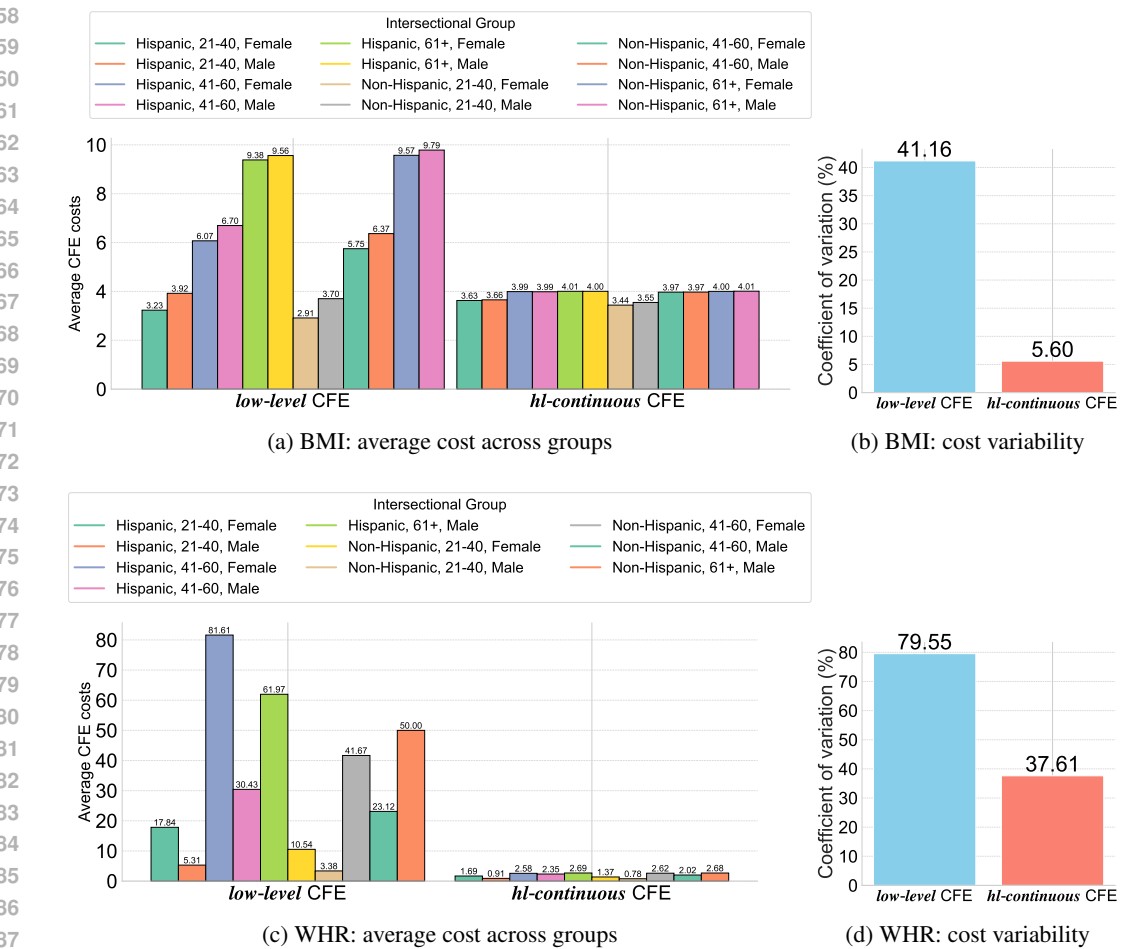

(a) BMI: average cost across groups

(b) BMI: cost variability

(c) WHR: average cost across groups

(d) WHR: cost variability

Figure 12: The figures illustrate the variations in the average costs incurred by individuals from different sensitive groups in BMI and WHR datasets when they take low-level or hl-continuous CFEs. Although not comparable across CFEs, (a) and (c) show the distribution of costs between groups within the CFE, and (b) and (d) show the coefficient of variations - indicating how variable around mean the average costs in groups are. Costs across sensitive groups vary more when individuals take low-level CFEs, than when they take hl-continuous CFEs.

### D.2.2 VARIED COSTS PREFERENCES

We model two types of hl-continuous CFEs: one where an hl-continuous action is in terms of Foods+monetary costs, and the other by Foods+caloric costs (see Appendix B.1.3). In a setting where negatively classified individuals care more about monetary costs over caloric costs, and vice versa, the CFE generator adapts to these diverse preferences and recommends the corresponding optimal CFE, as demonstrated in Figure 13.

Additionally, regardless of whether monetary or caloric costs were the desired costs by the individual, we consistently observed that hl-continuous CFEs involved fewer actions, resulted in more feature modifications and higher improvement (proximity) when compared to low-level CFEs (see Figure 11 and Figure 13 ).

Future research could investigate the data-driven CFE generation at the intersection of various settings. For instance, this could involve exploring Pareto-optimal solutions where individuals seek to simultaneously optimize multiple factors, such as monetary and caloric costs.

| Features to Change | Current Value | to | Required Value |
|---|---|---|---|
| Selenium (mcg) | 45 | → | 327.7319 |
| Total monounsaturated fatty acids (gm) | 12.392 | → | 89.34236700000017 |
| Total saturated fatty acids (gm) | 10.077 | → | 2.6004500000000004 |
| Vitamin B12 (mcg) | 1.21 | → | 0.1200000000000001 |
| Total folate (mcg) | 172 | → | 1179.7380000000003 |
| Vitamin B6 (mg) | 0.482 | → | 0.21794999999999998 |
| Niacin (mg) | 8.755 | → | 85.10721500000001 |
| Vitamin C (mg) | 35.7 | → | 0.10000000000000142 |
| Copper (mg) | 0.425 | → | 0.212950000000001 |
| Zinc (mg) | 2.61 | → | 1.3895 |
| Sodium (mg) | 1326 | → | 626.65 |
| Potassium (mg) | 994 | → | 6520.550000000004 |
| Phosphorus (mg) | 488 | → | 217 |
| Magnesium (mg) | 96 | → | 57 |
| Iron (mg) | 4.08 | → | 42.72946000000002 |
| Calcium (mg) | 309 | → | 113 |
| Total polyunsaturated fatty acids (gm) | 13.999 | → | 4.40896 |
| Carbohydrate (gm) | 109.45 | → | 43.376000000000005 |
| Dietary fiber (gm) | 4.1 | → | 50.113950000000024 |

(a) low-level CFE

| | action-1 | action-2 |
|---|---|---|
| Protein (gm) | 1.250 | 0.000 |
| Carbohydrate (gm) | 3.350 | 61.500 |
| Dietary fiber (gm) | 3.100 | 0.200 |
| Calcium (mg) | 52.000 | 8.000 |
| Iron (mg) | 0.830 | 3.720 |
| Magnesium (mg) | 15.000 | 2.000 |
| Phosphorus (mg) | 28.000 | 5.000 |
| Potassium (mg) | 314.000 | 16500.000 |
| Sodium (mg) | 22.000 | 52.000 |
| Zinc (mg) | 0.790 | 0.420 |
| Copper (mg) | 0.099 | 0.195 |
| Selenium (mcg) | 0.200 | 0.200 |
| Vitamin C (mg) | 6.500 | 0.000 |
| Niacin (mg) | 0.400 | 0.000 |
| Vitamin B6 (mg) | 0.020 | 0.000 |
| Total folate (mcg) | 142.000 | 0.000 |
| Vitamin B12 (mcg) | 0.000 | 0.000 |
| Total saturated fatty acids (gm) | 0.048 | 0.000 |
| Total monounsaturated fatty acids (gm) | 0.004 | 0.000 |
| Total polyunsaturated fatty acids (gm) | 0.087 | 0.000 |

(b) hl-continuous CFE with caloric costs

| | action-1 | action-2 |
|---|---|---|
| Protein (gm) | 0.000 | 19.440 |
| Carbohydrate (gm) | 61.500 | 0.000 |
| Dietary fiber (gm) | 0.200 | 0.000 |
| Calcium (mg) | 8.000 | 17.000 |
| Iron (mg) | 3.720 | 1.630 |
| Magnesium (mg) | 2.000 | 23.000 |
| Phosphorus (mg) | 5.000 | 139.000 |
| Potassium (mg) | 16500.000 | 179.000 |
| Sodium (mg) | 52.000 | 247.000 |
| Zinc (mg) | 0.420 | 0.690 |
| Copper (mg) | 0.195 | 0.050 |
| Selenium (mcg) | 0.200 | 70.600 |
| Vitamin C (mg) | 0.000 | 0.000 |
| Niacin (mg) | 0.000 | 10.136 |
| Vitamin B6 (mg) | 0.000 | 0.319 |
| Total folate (mcg) | 0.000 | 4.000 |
| Vitamin B12 (mcg) | 0.000 | 2.550 |
| Total saturated fatty acids (gm) | 0.000 | 0.211 |
| Total monounsaturated fatty acids (gm) | 0.000 | 0.107 |
| Total polyunsaturated fatty acids (gm) | 0.000 | 0.277 |

(c) hl-continuous CFE with monetary costs

Figure 13: When given actionable features values $[29.03, 109.45, 4.1, 309., 4.08, 96., 488., 994., 1326., 2.61, 0.425, 45., 35.7, 8.755, 0.482, 172., 1.21, 10.077, 12.392, 13.999]$, in the same order as shown in (b) and (c), for a negatively classified WHR individual, the low-level CFE generator recommends a CFE (a) with a cost of $56.588$. This CFE was unique to the individual. In contrast, the hl-continuous CFE generator generates two CFEs optimized for different individual's preferences. When optimizing for caloric cost, the CFE generator generates CFE (a) with a cost of $2.750$. This CFE, which was also optimal for other 25 negatively classified individuals, includes **action-1** (*consume endive, raw*) and **action-2** (*consume leavening agents: cream of tartar*). When optimizing for monetary cost, the CFE generator produces a CFE (b) of cost $4.010$. This CFE, also optimal for other 105 individuals, consists of **action-1** (*consume leavening agents: cream of tartar*) and **action-2** (*consume fish, tuna, light, canned in water, drained solids*). Lastly, while the low-level CFE (a) takes 19 actions, modifies 19 features and improves by 5679.95, the hl-continuous CFEs both take 2 actions, modify 19 features and improve by 16815.04 (b) and 16682.62 (c).

| Manual Groups | | Probabilistic Groups | |
|---|---|---|---|
| Group | Accuracy | Group | Accuracy |
| Group 0 | $0.881 \pm 0.01200$ | Group 0 (0.4) | $0.880 \pm 0.04400$ |
| Group 1 | $0.871 \pm 0.01260$ | Group 1 (0.5) | $0.771 \pm 0.02081$ |
| Group 2 | $0.875 \pm 0.01249$ | Group 2 (0.6) | $0.802 \pm 0.01571$ |
| Group 3 | $0.847 \pm 0.01359$ | Group 3 (0.7) | $0.873 \pm 0.01241$ |
| Group 4 | $0.886 \pm 0.01212$ | Group 4 (0.8) | $0.931 \pm 0.00947$ |

Table 3: Group-wise accuracy of the hl-discrete CFE generator on `manual groups` & `probabilistic groups` (see Appendix B.2.5).

### D.2.3 VARIED FEATURE SATISFIABILITY

In general, as shown in Table 2, compared to the unit threshold datasets: 20- 50- and 100-dimensional individual↦CFE datasets, individuals in the varied binary feature satisfiability datasets described in Appendix B.2.4 required fewer actions. This is mainly due to fewer number of features that individuals need to satisfy to get a desirable classification.

Our results show that without explicit knowledge of the varied feature satisfiability, when given test set individuals, the hl-discrete CFE generator trained on instances of a mixture of individual↦hl-discrete CFE varied feature satisfiability datasets successfully generate the right hl-discrete CFEs for the new individuals. The hl-discrete CFE generator achieves an accuracy of $99.683\%$ on `First10`, $99.496\%$ on `Last10`, $100\%$ on `First5`, $100\%$ on `Mid5`, and $100\%$ on `Last5`, dataset variants.

### D.2.4 VARIED ACCESS TO ACTIONS

The `Manual groups` individual↦hl-discrete CFE datasets (described in Appendix B.2.5) are more balanced in terms of the number of actions individuals take (see Figure 4(a)). The reason is individuals have access to the same distribution of hl-discrete actions, i.e., although individuals in each group have access to only a selected group of hl-discrete actions, all the hl-discrete actions for all groups were generated with the same probability, $p_a = 0.5$.

However, for the `Probabilistic groups` individual↦hl-discrete CFEs datasets (described in Appendix B.2.5), Figure 4(b) shows that as the probability of hl-discrete capabilities $p_a$ decreases, the number of hl-discrete individuals require to get all the necessary capabilities to transform their states to get a positive model outcome increases. In other words, individuals in certain groups only have access to more expensive and limited hl-discrete actions compared to others. For instance, individuals in the `Probabilistic groups` Group 0 face more difficulty (due to limited capabilities and more costly hl-discrete actions) in achieving positive classification outcomes than those in the Group 4.

Since the individuals in the `Manual groups` individual↦hl-discrete CFE datasets had more balanced access to hl-discrete actions as depicted in Figure 4(a), the hl-discrete CFE generators had almost similar accuracy ($\sim 87\%$) in the generation of CFEs across all individuals in different `Manual groups`, as shown in Table 3 (**left**). On the other hand, since the individuals in the `Probabilistic groups` had access to varied hl-discrete actions, the accuracy of the hl-discrete CFE generator varied greatly across the groups, as shown in Table 3 (**right**). For instance, as expected, the CFEs for `Probabilistic groups` Group 4 individuals with one-action hl-discrete CFEs were more accurately generated with an accuracy of $93.06\%$ as compared to Group 0 and Group 1 individuals, generated at an accuracy of $88.04\%$ and $77.09\%$, respectively.

### D.3 ACCURATE, CONFIDENT AND APPROXIMATE WHEN NEEDED

Our results show that the data-driven CFE generators are accurate and confident information-specific CFE generators. Additionally, unlike low-level CFE generators that sometimes fail to produce a CFE entirely for an individual, our data-driven CFE generators generate approximately good CFEs instead of no CFEs at all. The supplemental results in this appendix subsection are mainly for the fully-synthetic datasets.

### D.3.1 ACCURACY AND CONFIDENCE

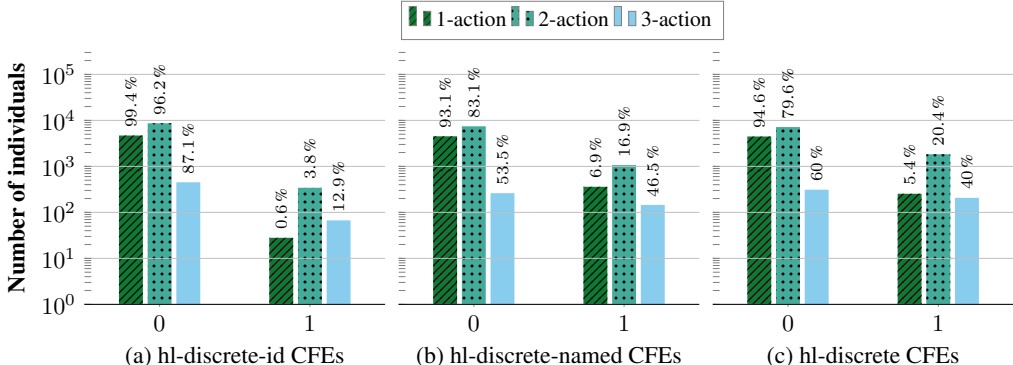

Figure 14: The data-driven hl-id CFE generator for the (**a**) hl-discrete-id CFEs, the hl-continuous CFE generator for the (**b**) hl-discrete-named CFEs, and the hl-discrete CFE generator for the (**c**) hl-discrete CFEs, achieved strong performance on the 20-dimensional `all` individual↦hl-discrete CFE, varied information access, test datasets (new individuals for the respective variants).

**Performance of the CFE generator on** 20**-,** 50**-, and** 100**-dimensional datasets**

|  | all | >10 | >40 |
|---|---|---|---|
| 20-dimensional | $0.969 \pm 0.00284$ | $0.984 \pm 0.00208$ | $0.993 \pm 0.00141$ |
| 50-dimensional | $0.744 \pm 0.00608$ | $0.838 \pm 0.00534$ | $0.915 \pm 0.00458$ |
| 100-dimensional | $0.354 \pm 0.00664$ | $0.630 \pm 0.00778$ | $0.856 \pm 0.00772$ |

Table 4: Accuracy of generation of hl-discrete-id CFEs for 20-dimensional, 50-dimensional and 100-dimensional: `all`, `>10`, and `>40` datasets.

The proposed data-driven CFE generators are evidenced to perform strongly on the varied datasets. As shown in Figure 14, on the 20-dimensional `all` individual↦CFEs dataset variants, the CFE generators achieved high accuracy at generating hl-discrete CFEs, hl-discrete-id CFEs, and hl-discrete-id CFEs. All the generators perform best on the single-action CFE individuals. Furthermore, with strong confidence, i.e., low margin error rates (see Table 4), the proposed data-driven CFE generators performed well on all datasets regardless of the data dimension or frequency of CFEs. Notably, they excelled on high-frequency datasets, that is to say, `>40` datasets regardless of the data dimensions, as seen in Table 4.

### D.3.2 APPROXIMATION

Unlike ILP-based low-level CFE generators, which do not generate CFEs for individuals when the ILP solution is sub-optimal or infeasible, our data-driven CFE generators alternatively produce valid CFE mistakes when suboptimal (see Figure 15), which might be preferable in retrospect. For example, of the $1.58\%, 16.23\%$ and $37.00\%$ mistakes the hl-id generator makes on the 20-, 50-, and 100-dimensional `>10` individual↦hl-discrete-id CFE datasets, $100\%, 99.23\%$, and $87.29\%$, respectively, were valid CFE mistakes. Similarly, the majority of the mistakes of the hl-discrete CFE generators were valid, e.g., on the 20-dimensional `>10` individual↦hl-discrete CFE dataset, of the $10.8\%$ mistakes the generator makes, $63.10\%$ were valid.

Additionally, the likelihood of the ILP-based low-level CFE generator's failure at generating CFEs (i.e., returns no CFEs) increases with the number of actionable features (data dimensions). Similarly, the percentage of valid mistakes from our proposed CFE generators decreases with the frequency of CFEs in the individual↦CFE training set, e.g., the percentage of valid mistakes is $87.29\%$ on the `>10` dataset and $57.83\%$ on the 100-dimensional `all` dataset.

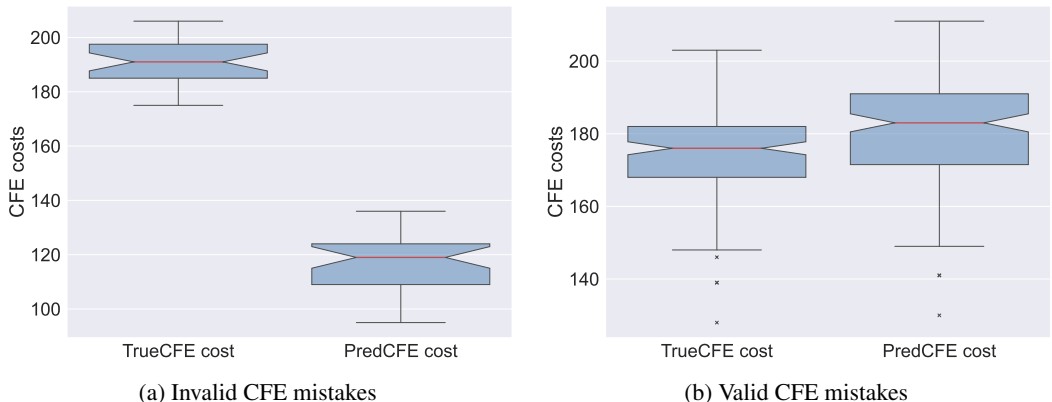

(a) Invalid CFE mistakes  (b) Valid CFE mistakes

Figure 15: A generated CFE is a mistake if the CFE doesn't match the true CFE. A valid CFE mistake transforms the individual's initial state to get a desirable model outcome. An invalid CFE mistake does not favorably transform the individual state. Distribution of costs of generated and true CFEs for (a) invalid and (b) valid CFE mistakes the hl-id CFE generator makes on 20-dimensional `all` individual↦hl-discrete-id dataset. Valid CFE mistakes are, by definition, more expensive than the true CFEs, while invalid CFE mistakes are cheaper than the true CFEs.

## D.4    EASIER TO SCALE AND MORE INTERPRETABLE

Our results demonstrate that our data-driven CFE generators—hl-continuous, hl-discrete, and hl-id—are more scalable than the low-level CFE generators. Furthermore, the costs and actions associated with the hl-continuous and hl-discrete CFEs are interpretable and more transparent, making them easier to validate and compare.

### D.4.1    SCALABILITY

Unlike the overly specific actions in the low-level CFEs (see Figure 13(a)), actions in hl-continuous and hl-discrete CFEs are more general, which allows to generalize the actions to various individuals. For example, our results, in Figure 16 show that while low-level CFEs were on average unique to a given individual, hl-continuous and hl-discrete CFEs were on average simultaneously optimal for several individuals (see Figure 16 and Figure 13).

Additionally, unlike the ILP-based low-level CFE generators that solve an expensive optimization problem for each new individual, our data-driven hl-continuous, hl-discrete, and hl-id CFE generators accurately and quickly generate CFEs without need for re-optimization.

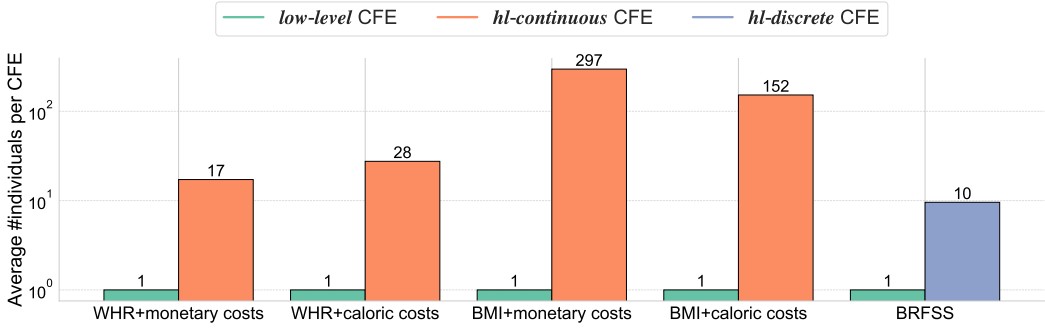

Figure 16: Regardless of the dataset considered, on average, while low-level CFEs were unique to a given individual, hl-continuous and hl-discrete CFEs were simultaneously optimal to multiple individuals.

### D.4.2 INTERPRETABILITY

The hl-continuous and hl-discrete CFEs consist of general, predefined actions, e.g., Figures 13(b) and 13(c) illustrates a typical hl-continuous: *take leavening agents: cream of tartar*. Due to this characteristic, these CFEs offer unique advantages over low-level CFEs, which are often overly specific and less straightforward for individuals to translate into practical actions (see Figure 13(a)). On the other hand, the hl-continuous and hl-discrete CFEs are more intuitive for users to interpret, execute, and compare with others. Additionally, the costs associated with these actions are comparable and easier to understand, with general knowledge of how they were derived—an essential factor for ensuring transparency in CFE generation.

### D.5 WORKS WELL WITH VARIOUS INFORMATION ACCESS CONSTRAINTS

With the purpose of investigating the effectiveness the data-driven CFE generators under various information access constraints, from the original individual↦hl-discrete CFE datasets, we created two more information access variants, individual↦hl-discrete-named CFE and individual↦hl-discrete-id CFE datasets as described in Appendix B.2.3. Given the individual↦hl-discrete CFE information access datasets, we use the data-driven hl-discrete CFE generators for the hl-discrete CFEs, hl-continuous CFE generators for hl-discrete-named CFEs, and hl-id CFE generators for hl-discrete-id CFEs.

In general, all the data-driven CFE generators, regardless of information access constraints described in Appendix B.2.3, generate single-action CFEs more accurately than multiple-action CFEs. For example, the hl-discrete CFE generator, as seen in Figure 14 (**c**), generates one-action CFEs at an accuracy of $94.6\%$, two-action CFEs at an accuracy of $79.6\%$, and three-action CFEs at an accuracy of $60.0\%$.

| | **Performance of CFE generators on** $20$**-dimensional datasets** | | |
|---|---|---|---|
| | all | >10 | >40 |
| hl-id CFE generator | $0.969 \pm 0.00284$ | $0.984 \pm 0.00208$ | $0.993 \pm 0.00141$ |
| hl-continuous CFE generator | $0.854 \pm 0.00581$ | $0.886 \pm 0.00531$ | $0.940 \pm 0.00411$ |
| hl-discrete CFE generator | $0.839 \pm 0.00605$ | $0.892 \pm 0.00518$ | $0.937 \pm 0.00420$ |

Table 5: Accuracy of CFE generators on 20-dimensional: all, >10, and >40 datasets.

However, in general, hl-id CFE generators were shown in Figure 14 (**a**) to 14 (**c**) and Table 5 to be more accurate and need less CFE frequency in the training set than the hl-continuous and hl-discrete CFE generators. For example, on the 20-dimensional all dataset, the hl-id CFE generator had an accuracy of $96.9\%$, compared to $85.4\%$ with hl-continuous CFE generator and $83.9\%$ with hl-discrete CFE generator.

### D.6 POTENTIAL CHALLENGES AND SOLUTIONS

We recognize several challenges faced by the proposed data-driven CFE generators: the low frequency of CFEs, the high number of actionable features, and the heavy reliance on the complexity of the CFE generator model. In this work, we thoroughly examine these challenges, propose plausible solutions, and suggest avenues for future research to explore these issues in greater depth.

#### D.6.1 NEGATIVELY AFFECTED BY HIGH NUMBER OF ACTIONABLE FEATURES

As the data dimensions (number of actionable features) increase, the number of actions individuals need to take also increases. For example, $54.4\%$ of the individuals in the 100-dimensional individual↦hl-discrete CFE dataset needed three hl-discrete actions and $0.0\%$ needed one hl-discrete (see Table 2). In comparison, $33.3\%$ of the individuals in the 20-dimensional individual↦hl-discrete CFE dataset had one action in their CFE, and very few, only $3.6\%$ of individuals had three actions in their hl-discrete CFEs (see Table 2).

In addition to an increase in actions needed, the uniqueness of CFEs also increases as the data dimension or the number of actionable features increases. The average frequency of the CFEs

for the `all` individual↦hl-discrete CFE training set for the 20-, 50-,and 100-dimensional datasets was $46.64\%, 21.75\%$, and $8.09\%$, respectively. Additionally, $18.115\%, 20.797\%$, and $31.072\%$ of the CFEs 20-, 50-, and 100-dimensional `all` individual↦hl-discrete CFE training sets, respectively, had a frequency of one (unique to one individual). Due to the low frequency of CFEs in the individual↦CFE datasets, after the train/test splits, some CFEs appeared in one data split and not the other. For example, for the 20-, 50-, and 100-dimensional `all` individual↦hl-discrete CFE datasets, there were $52, 154$ and $708$ unique CFEs in the test set not present in the training set, for the varied dimensional datasets respectively.

As a result, the data-driven CFE generators become less accurate as data dimensions increase. As seen in Table 4, in all cases, the hl-id CFE generator had the lowest accuracy on the 100-dimensional dataset and the highest on the 20-dimensional dataset. For example, while the hl-id CFE generator had an accuracy of $74.4\%$ on the 50-dimensional `all` dataset, it had an accuracy of $96.9\%$ on the 20-dimensional `all` dataset.

### D.6.2 NEGATIVELY AFFECTED BY LOW FREQUENCY OF CFEs

We created the varied frequency datasets: `all`, `>10`, and `>40` (see Appendix B.2.2) to study the effect of frequency of CFEs in the individual↦hl-discrete CFE dataset on the robustness of the data-driven CFE generators. After the train/test split, a frequency of atleast 20 individuals with the same CFE in the training set was insured with (`>40`) dataset. By definition, the `>40` datasets had the highest frequency of CFEs and `all` had the lowest. This frequency was also affected by data dimensions, as illustrated in Appendix D.6.1.

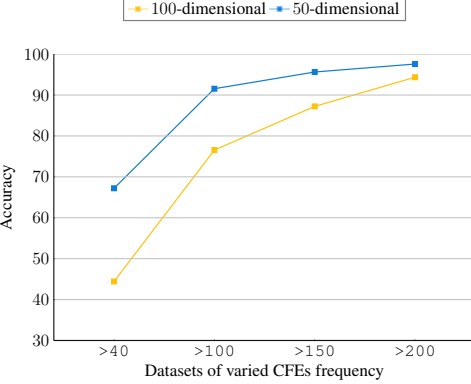

Figure 17: Accuracy of CFE generators improve with increase in the frequency of CFEs.

The low frequency of CFEs in the individual↦hl-discrete CFE training sets negatively impacted CFE generation across all datasets, regardless of data dimensionality. However, this effect became more pronounced as data dimensions increased. For instance, as shown in Table 4, the accuracy of CFE generators on the 20-dimensional dataset was highest when CFEs had a frequency of at least 20 in the training set (`>40`) and lowest on the `all` dataset, where some CFEs appeared in the test set but not in the training set. Specifically, the hl-id CFE generator achieved an accuracy of $99.3\%$ on the 20-dimensional `>40` dataset, compared to $96.9\%$ on the 20-dimensional `all` dataset. In contrast, CFE generation accuracy on the 20-dimensional dataset was significantly higher than on the 100-dimensional dataset. This difference highlights that the negative impact of low CFE frequency in the training set becomes more severe as data dimensionality increases.

Additionally, the minimum frequency of CFEs required for a strong CFE generator increases with number of actionable features. While the frequency of at least 20 in the training set ensured an accuracy of $99.3\%$ of the CFE generator on the 20-dimensional dataset (see Table 4), a higher frequency is needed for the 50- and 100-dimensional datasets (see Table 4 and Figure 17).

---

**Algorithm 1:** The individual$\mapsto$hl-discrete CFE dataset augmentation

> **Input:** an individual $\mathbf{x}$ and their hl-discrete CFE $I$, and the threshold classifier $\mathbf{t}$
> **Output:** valid derived augmentations of individual $\mathbf{x}$, $\mathbf{x}_{\text{augs}}$ with the same CFE
> **Data:** indices of features $ids$ where the hl-discrete CFE when taken, adds more than
>     needed capabilities to $\mathbf{x}$
> augs $\leftarrow 2^{|ids|}$ possible worse-off individuals;
> **foreach** aug *in augs* **do**
>   **if** aug *is valid* **then**
>     $\mathbf{x}_{\text{augs}} \leftarrow \mathbf{x}_{\text{augs}} \cup \{\mathbf{aug}\}$;
>   **end**
> **end**

---

**Data augmentation algorithm** We investigate the effect of increasing the frequency of CFEs, through data augmentation, on the performance of the data-driven CFE generator. The data augmentation algorithm described in Algorithm 1 is specific for the individual$\mapsto$hl-discrete CFEs datasets and can be generalized to other -hl-discrete CFEs generated with other threshold classifiers. To generate new individuals for which a given hl-discrete CFE is the most optimal, we ensure that no other hl-discrete CFE within the set of all hl-discrete CFEs can, at a lower cost, transform the new individual augment.

Therefore, given an individual state, we find all possible worse-off individual states such that the current optimal hl-discrete CFE is still the best CFE for the worse-off individual states. Worse-off individual states are those such that the features where the hl-discrete CFE is adding more capabilities than required to transform the individual state favorably are made worse, i.e., for $i$ such that $x_i^\star > t_i, aug_i < x_i$. Specific to the threshold classifier we use in the experiments, an hl-discrete CFE is adding more capabilities than required to feature $i$ of $\mathbf{x}$, if by taking the action, the transformed feature $x_i^\star$ is such that $x_i^\star > t_i$. The derived worse-off individual state (augment) $\mathbf{aug}$ is valid if $\mathbf{x}$'s hl-discrete CFE is also its the optimal CFE.

**Data augmentation reduces negative impact of low frequency of CFEs** With Algorithm 1, we augment the individual$\mapsto$hl-discrete CFE training set to increase diversity (AG1) and the frequency (AG2) of CFEs whose current frequency is less than 20 hl-discrete CFEs. For example, we reduce the number of hl-discrete CFEs with less than 20 individuals from 813 to 638, 2676 to 2005, and 9043 to 7144 for the 20- 50- and 100-dimensional datasets, respectively.

Experimental results show an improvement in the accuracy of the CFE generators on the test samples. For example, on the 100-dimensional dataset, the accuracy of the hl-id CFE generator increases from 35.37% before data augmentation to 50.54 after AG1, and 78.99% after AG2 (refer to Table 6). We, therefore, believe that data augmentation and other similar methods can be employed to improve the robustness of CFE generators in cases where there is a low frequency of CFEs in the individual$\mapsto$CFE training datasets.

| | **Effect of data augmentation** | | |
| --- | --- | --- | --- |
| | 20-dimensional | 50-dimensional | 100-dimensional |
| Before data augmentation | $0.969 \pm 0.00284$ | $0.744 \pm 0.00608$ | $0.354 \pm 0.00664$ |
| After AG1 | $0.965 \pm 0.00303$ | $0.760 \pm 0.00595$ | $0.505 \pm 0.00694$ |
| After AG2 | $0.982 \pm 0.00218$ | $0.845 \pm 0.00504$ | $0.790 \pm 0.00565$ |

Table 6: Data augmentation alleviates the negative effects of low frequency of CFEs and improves accuracy of data-driven CFE generators on the 20-, 50-, and 100-dimensional: `all` datasets.

### D.7 HEAVILY DEPENDS ON COMPLEXITY OF CFE GENERATOR MODEL

Given the individual$\mapsto$hl-discrete-id CFE 20-dimensional, `>40` dataset variant, we compare the effectiveness of the neural network-based CFE generator against the Hamming distance-based CFE generator. As shown in Figure 18, the neural network-based CFE generator demonstrates greater

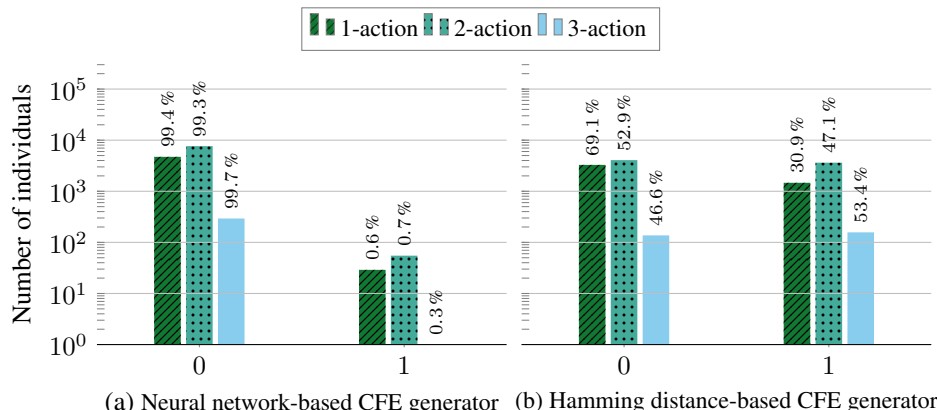

(a) Neural network-based CFE generator    (b) Hamming distance-based CFE generator

Figure 18: A comparison of accuracy of two CFEs generators on the 20-dimensional $>40$ dataset.

accuracy in generating CFEs for new individuals. Interesting for future works is an exploration of the effectiveness of CFE generators based on more advanced and alternative methods, e.g., multi-chain neural networks, reinforcement learning, and transformer models.

