# OpenReview forum: "Learning Actionable Counterfactual Explanations in Large State Spaces"
_ICLR.cc/2025/Conference — Submitted to ICLR 2025_

### Official Review · Reviewer_cP95 · 2024-10-22

**Soundness:** 2
**Presentation:** 3
**Contribution:** 2
**Rating:** 5
**Confidence:** 4

**Summary:**

The authors identify limitations in traditional low-level counterfactual explanation (CFE) generators and propose an alternative approach involving three data-driven counterfactual generators. These generators are trained with high-level CFEs, which consist of combinations of high-level actions, where each action can alter multiple features. The high-level counterfactuals are defined as follows: (1) hl-continuous CFE, representing the lowest-cost subset of hl-continuous actions that can achieve recourse (either by addition or subtraction of the feature changes that correspond to each action); (2) hl-discrete CFE, which is formulated as a minimum weight set cover problem using hl-discrete actions (limited to the positive feature changes that correspond to each action); and (3) hl-id, for which the exact cost and the resulting feature changes are unknown. For (1) and (2) once the counterfactuals are defined, an integer linear programming (ILP) solution is applied to each instance, creating a training dataset consisting of (X, CFE) pairs. The generators are trained with these datasets.

**Strengths:**

1. Once the generator is trained, the counterfactual explanation (CFE) prediction is very efficient. With proper training and a high-quality dataset, the generator can achieve high accuracy in its results.
2. If a predefined set of actions is available, generating counterfactuals for subgroups can be more straightforward.
3. The proposed approach enables fairness auditing in a more structured and systematic manner.

**Weaknesses:**

1. Training the generator requires solving integer linear programming (ILP) problems, which are specifically suited for linear classifiers. For other types of classifiers, creating the dataset necessitates brute-force methods, which may not be efficient.
2. The hl-discrete CFEs are limited to positive changes, which may not apply to non-linear models. Even for linear models, this approach requires additional preprocessing and a deeper understanding of the model's internal.
3. The hl-continuous and hl-discrete generators still rely on query access to the model and the cost of actions to create the training set, meaning that these requirements are not fully eliminated.
4. For the hl-id generator, the challenge remains in how to construct the training dataset, as the process is not clearly defined.

**Questions:**

1. Consider introducing the notion of high-level actions at the beginning of the introduction, defining them as combinations of low-level actions.
2. Why are hl-continuous counterfactuals considered superior to low-level counterfactuals? While having a predefined set of high-level actions with known outcomes can simplify the CFE generation process and facilitate testing, the feasibility of combining actions may not always be straightforward, and the constraints could be more complex. Have you tested your approach on more complex datasets with additional constraints to demonstrate its effectiveness?
3. For the ILP in Definition 2 (line 196), the condition $\epsilon_j\leq a_j$ appears to be redundant. If $a_j=0$ then $e_j$ has no effect, and if $a_j=1$ this condition always holds since both $\epsilon_j$ and $a_j$ are in $\{0,1\}$. Additionally, there seems to be a typo on the same line, where $\epsilon_i$ should be $\epsilon_j$.
4. Could you clarify the need for hl-discrete actions? It seems unnatural to limit changes to only "positive changes." Moreover, since the focus is on binary features, isn't this approach overly restrictive in your experiments, where $t = 1$ means targeting a specific state?
5. For the ILP in Definition 4, the constraint on line 220 should be written as $\sum_{j\in J}d_{ji}a_j +x_{i} \geq t_i$.
6. Could you clarify how the generators produce "generalizable CFEs" (line 52)
7. Regarding Figure 2 (a) Comparing actions may not be meaningful, as a high-level action can modify multiple features simultaneously, whereas a low-level action changes only one feature (also stated on line 356). (b) In the bibliography, the number of features changed is important for explainability. You should emphasize that, in the case of high-level actions, having more feature changes does not necessarily reduce explainability. (c) a more distant state does not necessarily indicate a higher improvement. Since most ML models used for generating CFEs are non-linear, the concept of CFEs is inherently linked to minimizing cost. Thus, the goal of a CFE generator should not be maximizing improvement, which contradicts the problem being addressed. (d) Higher frequency is expected because the actions are predefined, resulting in more "global" CFEs. (General comment:) It would be beneficial to include variance or standard deviation in the figures for a more comprehensive analysis.
8. In line 367, you mention "diverse improvements." Could you clarify what you mean by this?

---

> ### Author Response · Authors · 2024-11-27
> **Response to the reviewer cP95**
>
> Thank you reviewer cP95 for taking time to review our work. Also, thank you so much for the constructive feedback, we have edited the PDF within page limits to address most of the issues you raise.  Below are responses to questions you raised.
>
> > Training the generator requires solving integer linear programming (ILP) problems, which are specifically suited for linear classifiers. For other types of classifiers, creating the dataset necessitates brute-force methods, which may not be efficient.
>
> The hl-discrete CFEs are limited to positive changes, which may not apply to non-linear models. Even for linear models, this approach requires additional preprocessing and a deeper understanding of the model's internal.
> The reviewer is right that in the non-data-driven algorithmic setting, the CFE generation is limited and nontrivial. We propose data-driven CFE generators that, once trained on the individual-CFE dataset, the CFE generator can generate CFEs without re-optimization or running brute-force methods for that specific new (test) example.
>
>
> > The hl-continuous and hl-discrete generators still rely on query access to the model and the cost of actions to create the training set, meaning that these requirements are not fully eliminated.
>
> When generated in a non-data-driven algorithmic way, for example by using the ILPs (equations 2 and 3), we need query access to the classifier. However, in settings where the CFE generation is data-driven, there will be no need for query access to the model. Our proposed data-driven CFE generators for hl-discrete, hl-continuous, and hl-id CFEs eliminate the need for those requirements.
>
> > Consider introducing the notion of high-level actions at the beginning of the introduction, defining them as combinations of low-level actions.
>
> We agree with the reviewer that in some sense, the high-level CFEs are a "combination" of low-level CFEs. However, the term doesn't accurately capture the relationship between these CFEs. In a more general sense, it could be viewed as low-level CFEs being more specific and high-level CFEs more general and user-friendly.
> For example, although high-level actions, e.g. hl-continuous actions change features in ways low-level actions do, high-level actions are not necessarily a combination of low-level actions. Additionally, because low-level actions heavily depend on the action set, defined based on the feature space of the training data, they could be limited in ways the high-level actions might not be because they are predefined and might transcend the limited classification training data.
>
> >Why are hl-continuous counterfactuals considered superior to low-level counterfactuals? While having a predefined set of high-level actions with known outcomes can simplify the CFE generation process and facilitate testing, the feasibility of combining actions may not always be straightforward, and the constraints could be more complex. Have you tested your approach on more complex datasets with additional constraints to demonstrate its effectiveness?
>
> The hl-continuous actions in addition to having advantages the reviewer mentions, are also closer to the final implementable step than low-level actions (see for example figure 1a). Additionally, with fewer actions, the individual changes more features and achieves higher improvement (recourse).
>
> In cases where there is limited information on the effect of hl-continuous actions on the features, it would be hard to solve the ILP. Investigating this use case would be interesting for future works.
> In our work, we also consider cases where the decision-maker has access to individual-CFE datasets, which makes it easier to generate CFEs with even more limited information. In this case, the scenario above would be less of a challenge.

---

> > ### Comment · Reviewer_cP95 · 2024-11-30
> > **Reviewer response to authors**
> >
> > Thank you for your response and your effort to answer my questions. However, I need additional information.
> >
> > Regarding your first two answers. Indeed in a data-driven setting, this is not an issue. But in order to have a data-driven setting you need the data, that normally are not available and need to be generated somehow. How will you get the information you need? The hl-discrete counterfactuals e.g., are defined as the solution of the ILP 2. Without this information how will you create such a dataset? Also in the case of a more complex model, I do not see how your definitions should be edited to make sense.
> >
> > >  Additionally, because low-level actions heavily depend on the action set, defined based on the feature space of the training data, they could be limited in ways the high-level actions might not be because they are predefined and might transcend the limited classification training data.
> >
> > I do not understand this. How do low-level actions depend on the action set while high-level actions do not? How do you define the action set in the first case? Also, high-level actions, in the same way, can be limited since they allow specific changes.
> >
> > > Why are hl-continuous counterfactuals considered superior to low-level counterfactuals? While having a predefined set of high-level actions with known outcomes can simplify the CFE generation process and facilitate testing, the feasibility of combining actions may not always be straightforward, and the constraints could be more complex. Have you tested your approach on more complex datasets with additional constraints to demonstrate its effectiveness?
> >
> > You didn't answer my question. For example, in the ILP of problem 2, the solution for any individual is a combination of high-level actions. Isn't this a very "simplified" setting, without any feasibility constraints? It is not clear to me if you have considered e.g., some datasets made from a non-linear setting.

---

> ### Author Response · Authors · 2024-11-27
>
> > Could you clarify the need for hl-discrete actions? It seems unnatural to limit changes to only "positive changes." Moreover, since the focus is on binary features, isn't this approach overly restrictive in your experiments, which means targeting a specific state?
>
> The discrete actions, especially the ones we define (binary) would be useful for cases where features can be thresholded, as in the case we consider, health wellness. We agree that in the current setup, it's quite restricted. There are several interesting future directions, for example
> considering actions whose effects are stochastic.
>
> > Could you clarify how the generators produce "generalizable CFEs" (line 52)
>
> This term is about how easy it is to apply a CFE to several individuals, which is not the case with low-level CFEs. That is, the low-level CFE is very specific in a way that makes it unique to one individual, and this could be explained by the low-level CFE generator optimization function. Experimentally, we also observed a perfect 1:1 relationship between the number of actions taken and the number of features modified in low-level CFEs, which wasn't the case for high-level CFEs (figure 9b appendix). Figure 2a) and Figure 16 (appendix) show that in general, a high-level CFE was on average applied to several individuals.
>
> > Regarding Figure 2 (a) Comparing actions may not be meaningful, as a high-level action can modify multiple features simultaneously, whereas a low-level action changes only one feature (also stated on line 356). (b) In the bibliography, the number of features changed is important for explainability. You should emphasize that, in the case of high-level actions, having more feature changes does not necessarily reduce explainability.
>
> This is true, with high-level CFEs, changing more features does not reduce explainability as it does in the low-level CFEs. Additionally, the number of features modified does not necessarily imply more actions have to be taken in high-level CFEs as is the case with low-level CFEs.
>
> (c) a more distant state does not necessarily indicate a higher improvement. Since most ML models used for generating CFEs are non-linear, the concept of CFEs is inherently linked to minimizing cost. Thus, the goal of a CFE generator should not be maximizing improvement, which contradicts the problem being addressed.
>
> Indeed, distant changes might not always indicate the highest improvement. In our work, we only looked at improvements as a function of the l2 distance between the initial state and the new state after the agent takes recommended actions. It would be interesting for future works to investigate other aspects or dimensions of improvement.
> Minimizing cost is indeed primal to CFE generation. We do however note in our work that linking cost directly to features changed and improvements made might not be inaccurate. When cost is attached to predefined actions, and not directly to features changed and by what amount, the cost is in some sense not sensitive to improvement as it would be in a direct case. Our argument is not that we are directly optimizing for maximum improvement, but rather, that using predefined actions leads to higher improvement than low-level actions. It's an indirect advantage we observed with the proposed CFE generation approaches.
>
> >(d) Higher frequency is expected because the actions are predefined, resulting in more "global" CFEs. (General comment:) It would be beneficial to include variance or standard deviation in the figures for a more comprehensive analysis.
>
> Because the nature of the high-level CFEs and low-level CFEs differed, we didn't use standard deviation, but rather compared the coefficient of variations for the two settings, especially in cases where we wanted to directly compare the distributions of the characteristics of the two CFEs.
>
>
> >In line 367, you mention "diverse improvements." Could you clarify what you mean by this?
>
> This was meant in the sense that because the predefined high-level actions are often diverse - that is to say, modify multiple features simultaneously - the high-level CFEs led to diverse improvements, even though that was not directly optimized for. This can be seen in Figure 2a) and Figure 8) where high-level CFEs simultaneously modify multiple features and higher improvement (recourse).
>
> *Please let us know in case of more questions or clarifications, and thank you so much again for taking time to engage with us!*

---

> > ### Comment · Reviewer_cP95 · 2024-11-30
> > **Reviewer response to authors**
> >
> > > This term is about how easy it is to apply a CFE to several individuals, which is not the case with low-level CFEs. That is, the low-level CFE is very specific in a way that makes it unique to one individual, and this could be explained by the low-level CFE generator optimization function. Experimentally, we also observed a perfect 1:1 relationship between the number of actions taken and the number of features modified in low-level CFEs, which wasn't the case for high-level CFEs (figure 9b appendix). Figure 2a) and Figure 16 (appendix) show that in general, a high-level CFE was on average applied to several individuals.
> >
> > Since the setting is data-driven and the actions are predefined, how can you claim that the CFEs are generalizable? Also, the 1:1 relationship has to do with the way you defined low-level CFEs. In general, CFEs are not bound to one-feature change.
> >
> > > Indeed, distant changes might not always indicate the highest improvement. In our work, we only looked at improvements as a function of the l2 distance between the initial state and the new state after the agent takes recommended actions. It would be interesting for future works to investigate other aspects or dimensions of improvement. Minimizing cost is indeed primal to CFE generation. We do however note in our work that linking cost directly to features changed and improvements made might not be inaccurate. When cost is attached to predefined actions, and not directly to features changed and by what amount, the cost is in some sense not sensitive to improvement as it would be in a direct case. Our argument is not that we are directly optimizing for maximum improvement, but rather, that using predefined actions leads to higher improvement than low-level actions. It's an indirect advantage we observed with the proposed CFE generation approaches.
> >
> > Can you claim this in any model that does not have linear bounds? CFEs goal in general is about the minimum cost to achieve the desired outcome. From my understanding, in your case, higher improvement means higher cost/more distant points, which contradicts the CFEs notion. So how are high-level counterfactuals better solutions to CFEs than low-level counterfactuals?
> >
> > > This was meant in the sense that because the predefined high-level actions are often diverse - that is to say, modify multiple features simultaneously - the high-level CFEs led to diverse improvements, even though that was not directly optimized. This can be seen in Figure 2a) and Figure 8) where high-level CFEs simultaneously modify multiple features and higher improvement (recourse).
> >
> > Regarding diversity, you connect diverse CFEs with diverse improvements. How do you define diversity in the "improvement"?
> > Talking about diverse CFEs, do you classify [1] as high or low-level actions?
> > You also somehow make a connection to the higher improvement with recourse. Achieving recourse means that you change your situation to achieve a desired output e.g. classified in the positive class. The improvement is not correlated with the notion of recourse. Mainly we connect cost with how you achieve recourse, which is your goal to minimize in the "individual case".
> >
> > [1] Explaining machine learning classifiers through diverse counterfactual explanations

---

> ### Author Response · Authors · 2024-12-02
>
> Thank you reviewer cP95 for your feedback and detailed follow-up questions!
>
>
> > Regarding your first two answers. Indeed in a data-driven setting, this is not an issue. But in order to have a data-driven setting you need the data that normally are not available and need to be generated somehow. How will you get the information you need? The hl-discrete counterfactuals e.g., are defined as the solution of the ILP 2. Without this information how will you create such a dataset? Also in the case of a more complex model, I do not see how your definitions should be edited to make sense.
>
> We agree with the reviewer that one needs access to individual-CFE datasets. For the results in our paper, we generate these datasets by solving ILPs (Equations 2 and 3). In practice,  individual-CFE data is available at scale in domains that include healthcare, education (e.g., college admissions), banking (e.g., loan decisions), and legal settings (e.g., judicial decisions). However, data from such domains may not be publicly available (e.g., due to privacy constraints). As we discussed in Section 6, federated learning and secure multi-party computation would provide a viable means of training our models.
>
> However, we highlight the advantage of our method, specifically in settings where the party that generates CFEs may not have access to some information, e.g., access to the classifier. Our proposed data-driven CFE generation method has the advantage that it provides a mechanism for producing CFEs despite these challenges.
> Consider a scenario in which the decision-making party is not exactly the same as the one generating the CFE, as can be the case in practice. Take for instance a bank management and loan officers. Loan officers can have access to a trained data-driven CFE generator, without requiring information about (or access to) the classifier or the data on which the classifier was trained. In this setting, it is sufficient if only a limited group (e.g., bank management) has access to the data used to create the CFE generator. Therefore, a  trained CFE generator model could easily be transferred to (i.e., shared with) relevant parties to be used for generating CFEs for future new instances.
> The same can not be said about low-level CFE generators or the individual-based CFE generators, because loan officers with less access privileges within the bank management setup will not be able to generate good CFEs for clients.
>
> > I do not understand this. How do low-level actions depend on the action set while high-level actions do not? How do you define the action set in the first case? Also, high-level actions, in the same way, can be limited since they allow specific changes.
>
> We apologize for the confusion. Both depend on the action set—our rebuttal sentence was overloaded. We were specifically referring to an action set that is collection of specialized action elements for each feature and whose definition depends on the training set and the choice of various settings' choices: e.g., for mutability, stepsize, steptype, among others (for example, see Ustun et al. (2019) and the example [here](https://github.com/ustunb/actionable-recourse/blob/master/examples/cfpb_tech_sprint_demo.ipynb)), whereas in our setting, the action set is based on the predefined actions.
> The reviewer is correct in that high-level actions might be limited since they allow specific changes, but they are not  limited by training data and manual definition actionability parameters which could be subject to errors and bias. Granted, our proposed method does not solve all issues, but it does, in our opinions, offer valuable improvements to the CFE generation process.

---

> ### Author Response · Authors · 2024-12-02
>
> > You didn't answer my question. For example, in the ILP of problem 2, the solution for any individual is a combination of high-level actions. Isn't this a very "simplified" setting, without any feasibility constraints? It is not clear to me if you have considered e.g., some datasets made from a non-linear setting.
>
> It’s true that the feasibility of combining actions may not always be straightforward, and the constraints could be more complex. We do think that this challenge is also implicit in the generation of low-level CFEs. For example, it might not necessarily be straightforward to know if the two actions: 1) increase calcium(mg) from 309 to 113 and 2) increase zinc(mg) from 2.61 to 1.3895 can be feasibly combined (i.e., they may not be independent actions), and if actually cheaper than alternative changes (CFEs), as announced. In fact, with low-level CFEs individuals might have to incur extra costs to translate low-level actions, e.g., “increase  zinc(mg) from 2.61 to 1.3895” into action that can be implemented in practice.
> Although high-level CFEs improve the communication and interpretability of recourse to individuals, we acknowledge that our approach comes with limitations and does not solve all CFE generation challenges, and we think the reviewer raises an interesting problem - feasibility of combining actions -  to explore for future work!
>
> > Since the setting is data-driven and the actions are predefined, how can you claim that the CFEs are generalizable? Also, the 1:1 relationship has to do with the way you define low-level CFEs. In general, CFEs are not bound to one-feature change.
>
> We agree that how you define the low-level CFEs definitely has an effect and is not always bound to one feature change. However, for low-level CFEs, we do observe an almost perfect relationship between number of  actions taken and number of features modified .
> Additionally, consider the setting where one might want to generalize the generation of low-level CFEs using a data-driven approach. We note that unlike high-level CFEs, low-level CFEs are overly specific (see for example figure 1a) and consequently very unique to individuals. Therefore because of this, it makes it extremely hard and expensive to move from ILP or individual based low-level CFE generation to a data-driven approach, while this is a relatively natural step to take with high-level CFEs.
>
> > Can you claim this in any model that does not have linear bounds? CFEs goal in general is about the minimum cost to achieve the desired outcome. From my understanding, in your case, higher improvement means higher cost/more distant points, which contradicts the CFEs notion. So how are high-level counterfactuals better solutions to CFEs than low-level counterfactuals?
>
> We mean higher improvement *in comparison* to the low-level CFEs.  We still find the cheapest (least) set of actions that enable the individual to get a desirable outcome.
> However due to the difference in nature of action sets, high-level CFEs typically lead to higher improvement/recourse.
> Additionally, the costs associated with low-level CFEs might actually not translate into real-world costs, and more costly than anticipated for example, costs might be incurred to translate low-level CFEs to implementable steps. See for example “*features cannot be made commensurate by looking only at the distribution of the training data*” paragraph, among others in Barocas et al., 2020.
>
> > Regarding diversity, you connect diverse CFEs with diverse improvements. How do you define diversity in the "improvement"? Talking about diverse CFEs, do you classify [1] as high or low-level actions? You also somehow make a connection to the higher improvement with recourse. Achieving recourse means that you change your situation to achieve a desired output e.g. classified in the positive class. The improvement is not correlated with the notion of recourse. Mainly we connect cost with how you achieve recourse, which is your goal to minimize in the "individual case".  [1] Explaining machine learning classifiers through diverse counterfactual explanations
>
> We define diversity in terms of a variety and number of features changed. In comparison to low-level CFEs, high-level CFEs tend to change on average a higher number of features, without adding extra constraints to the optimization problem.
> It’s true that improvement is not always correlated with the notion of recourse. In our case, since the actions are predefined which  increases this correlation, unlike the overly specific and tied to features kind of actions, as is the case with low-level CFEs, e.g., calcium(mg) from 309 to 113 which opens this to being interpreted by individuals in various ways and achieved (through gaming/improvement) however way they want.

---

### Official Review · Reviewer_iX5Z · 2024-11-02

**Soundness:** 1
**Presentation:** 1
**Contribution:** 1
**Rating:** 1
**Confidence:** 5

**Summary:**

The paper proposes to provide higher-level actions as CFEs to individuals who have received undesirable predictions from a classifier, instead of low-level CFEs that most previous techniques provide. The authors claim that higher-level CFEs are sparser, make more improvement to an individual, are diverse, more fair and personalized.

**Strengths:**

The paper proposes to tackle an interesting problem of providing higher level actions, that might make it easier to communicate actionable changes to a user

**Weaknesses:**

The paper suffers from several weaknesses. I have enumerated them below, and marked the ones that are major. If the authors like, they can only focus on the major weaknesses:

1. [Major] The paper's motivation states that their proposed technique will be useful in cases when an individual does not have access to the priviledged information such as the ability to query the classifier. Now the entire proposed technique hinges on the ability to generate CFEs for all individuals and then train another model on that top of that? So how is an individual who does have access to a classifier use this technique when the pre-requisite of the technique is to generate CFEs not for 1 just thousands of individuals and then use the technique to do something for that individual? The proposed technique does not need less privilege than having access to the classifier, in facts it requires a 1000X more priviledge, the ability to generate CFEs for thousands of individuals. So the question is how is your technique applicable for an individual who does have access to a classifier as claimed in your motivation?
Could the authors explain how their method would be deployed in a real-world setting where individual users lack access to the classifier? Who would be responsible for generating the initial dataset of CFEs, and how would this be done with the restrictions the paper aims to address?"

2. in line 152 you mention there are 4 notable limitations of low-level CFEs. There are no references to show that these limitations are indeed "notable".

3. In line 154 you say: solving this problem is computationally expensive as it is NP-hard optimization for each new agent. What is an agent? Please define it. Also in practice this optimization is quite cheap, fast, and scalable. See the dozens of the papers cited in Verma et al 2020.

4. In line 157, you say that "each action modifies an individual feature" -- that is not true for most causality based CFE generation techniques, like Consequence-aware sequential counterfactual generation, Geco: Quality counterfactual explanations in real time, Amortized generation of sequential algorithmic recourses for black-box models.

5. [Major] Given example of a situation where a CFE providing organization would not have access to the underlying classifier. Usually CFEs are provided for financial situations like loans and credit cards, and the banks are supposed to provide CFEs. In this case, the banks have access to their own models. Your current example of a dietician and celiac **does not** do a good job, celiac classification is a very simple rule based thing and the advice (CFE as you say) is known to all the physicians. No body uses a complex model that makes a decision that nobody understands (due to it being a black-box model) and therefore you need a CFE. Could the authors provide a more relevant example, perhaps from domains like algorithmic hiring or credit scoring, where the decision-making process is less transparent and the need for actionable explanations is more apparent?

6. [Major] Is Hl-discrete not the same as HL_continuous with just binary weights on the features? Is that is the case, then why do you make it as a separate case, its just a special case of HL_continuous, does not need that much attention and focus in the paper.

7. The HL-ID suggestions are generic and not useful, it is like a banker suggesting to a person whose loan request is canceled to increase their credit score or their bank balance -- everybody knows that. The point of a CFE is to provide a specific and small changes that are easily achievable, not generic advice.

8. [Major] Line 258 Please define what you mean by the symbol individual -> CFE dataset. From what I understand this is a dataset of individuals and their corresponding CFEs (which you never mention how you computed)

9. [Really Major] What is the job of the neural networks you train in section 4.2? From what I understand you first gather this dataset of individuals and CFEs, then somehow you get the actions for HL-continuous (mentioned on line 322-323) and then you use these actions to generate CFEs for new individuals? If this is the case, then what do you need the neural networks for, you can just use ILP to do this as stated in Section 3.1. The experimental section is really badly written and I did not understand the motivation of what you are doing at all. Please rewrite the section stating why you did something before telling what you did and even before that show the full pipeline of what you plan to do (ideally in a figure). Could the authors provide a clear, step-by-step explanation of their method's pipeline, including:

**a)** How the initial dataset of individuals and CFEs is generated
**b)** The specific role of neural networks in the process
**c)** How this approach differs from or improves upon using ILP directly
**d)** A diagram illustrating the full pipeline from data generation to CFE prediction for new individuals

10. [Major] Where do you magically get the actions to use in HL-continuous (or discrete) you mention in line 322?

11. Please explain the job of neural networks in HL-ID CFE generator? In this case you don't even have access to the classifier, so what loss are you optimizing over? You don't even know if the actions you propose will do anything to change the prediction of the classifier (because of the setting)

12. The assumptions and statements in the experimental results section 5.1 are either unsupported or obvious:
      1. Line 362: sparsity might be undesirable and challenging because individuals aim to implement as many changes as possible: No this is not true at all, in case of CFEs individuals want to just change the classifier prediction and want to implement as **less** changes as possible to get it. Your examples of health are invalid because nobody uses a complex neural network classifier to classify if someone is unhealthy and healthy, and your goal is not to change the prediction of the classifier, but to become healthy. And therefore in a health example you want to implement a lot of changes, not in the case of a usual CFE like a financial situation.

       2. Line 372: Despite hl-continuous CFEs having fewer actions on average, they result in more feature changes: This is **really obvious** in your case. You are just clubbing multiple changes to several features and calling that one action. I could define one action that changes all features and that final datapoint is classifier as positive by the classifier. So then I can say my proposed techniques just requires 1 action and this action changes all the features and achieves 100% success rate. I don't think that is useful.

      3. Line 404: Additionally, on average, hl-continuous CFEs, with fewer actions (∼ 2) result in states that are more distant: This is also **obvious**. If your actions make large changes in features, it is obvious that the new datapoint will be more distant.

       4. Line 409: hl-continuous CFEs tend to be more desirable for decision-makers and individuals alike: how can you claim this? Did you do a user study or even any automated metric to show this? I don't think this is true.

13. Line 390: What does "having a higher CFEs frequency" mean? What is CFEs frequency?

**Questions:**

All my questions are written in the weakness section. Overall, the authors really need to convince that their technique is usuable at all (because currently it seems it is not -- one needs to have access to thousands of CFEs to use this technique) and write the section on how to actually use the technique (section 3.2 and 4.1 need to be explained much more better with a motivation for each part)

---

> ### Author Response · Authors · 2024-11-27
> **Response to the reviewer iX5Z**
>
> Thank you Reviewer iX5Z for taking time to review our work. We have edited the paper in several ways (within the page limits) and below we address most of the questions, especially the ones the reviewer flagged as major, below.
>
> > So the question is how is your technique applicable to an individual who does have access to a classifier as claimed in your motivation?  Could the authors explain how their method would be deployed in a real-world setting where individual users lack access to the classifier? Who would be responsible for generating the initial dataset of CFEs,  and how would this be done with the restrictions the paper aims to address?
>
> The ILPs we defined (Equations 1 and 2) required access to a classifier/threshold function. The ILPs provide information on the composition of the hl-continuous and hl-discrete CFEs and their generation in a non-data-driven algorithmic way. Our proposed data-driven CFE generators that rely on the instances of individual-CFE datasets are computationally cheaper because they don't require solving an NP-hard problem for each individual. The CFE generators are trained once and used to generate CFEs for several new individuals.
>
> When the decision-maker has access to historical individual-CFE data, e.g., high-school counselors with historical student-college admission data, hospitals with patient-interventional data, among others, they can use this individual-CFE data to train a CFE generator to generate CFEs for new individuals. With the data-driven CFE generators, decision-makers don't need query access to the classifier.
>
> > Could the authors provide a more relevant example, perhaps from domains like algorithmic hiring or credit scoring, where the decision-making process is less transparent and the need for actionable explanations is more apparent?
>
> In a real-world setting when generating CFEs and recourse, access to the decision-making classifier is often a big challenge, especially because the people who provide recourse are not always the same as those who predict the individual’s class/target label. For example, using historical student to college admission data, high-school counselors who don't have access to college admission classifiers, often provide actionable advice to help students to get into colleges.
> The recourse generators often have access to historical individual-CFE (interventional) data and use this information to generate CFEs for new individuals.

---

> > ### Author Response · Authors · 2024-11-27
> >
> > > Where do you magically get the actions to use in HL-continuous (or discrete) you mention in line 322?..
> >
> > We didn’t fully understand what the reviewer meant by the term magical about this process. Section 4.1 describes the process by which we identify the hl-continuous and hl-discrete datasets (including how we identify the actions for each case). Appendices B.1 and B.2 provide additional details.
> >
> > > Line 258 Please define what you mean by the symbol individual -> CFE dataset. From what I understand this is a dataset of individuals and their corresponding CFEs (which you never mention how you computed)...
> >
> > The reviewer is correct—individual-CFE denotes individuals and their corresponding CFE. We mention how we compute these datasets, Section 4.1 describes how we obtain the real-world datasets (Lines 266–278), the semi-synthetic datasets (Lines 280–291); and the fully synthetic datasets (Lines 293–302). We provide further details in Appendices B.1 and B.2.
> >
> > >Is hl-discrete not the same as hl_continuous with just binary weights on the features? If that is the case, then why do you make it a separate case? It's just a special case of hl_continuous, and does not need that much attention and focus in the paper.
> >
> > Although, in some sense, hl-discrete CFEs can be viewed as more general cases of hl-continuous and low-level CFEs, the reverse is not true.
> > The discrete actions, especially the ones we define (binary) would be useful for cases where features can be thresholded, as in the case we consider, e.g., health wellness. There are also several interesting future directions specific to this case, e.g., considering fractional actions and those with stochastic effects on features.
> >
> >
> > >What is the job of the neural networks you train in section 4.2? …
> >
> > One of the main goals of our paper is to propose a generalizable, data-driven approach to CFE generation that overcomes the limitations of contemporary CFE generators, which are constrained to providing low-level explanations. We use neural networks as the underlying model for data-driven CFE generation. We also show how the complexity of the data-driven CFE generator affects performance.
> >
> >
> > >In line 152 you mention there are 4 notable limitations of low-level CFEs. There are no references to show that these limitations are indeed "notable".
> >
> > We described the limitations as notable because they are relatively known in the CFE and recourse community, as evidenced by their inclusion in several survey or position papers, e.g. in Barocas et al. (https://arxiv.org/pdf/1912.04930)). That said, we appreciate the reviewer’s suggestion that we corroborate these claims. We have updated the paper to include references to papers that discuss these limitations.
> >
> > >What is an agent?
> >
> > We use the term "agent" to refer to the individuals being classified and to whom explanations are provided as defined in Line 37 of Section 1.
> >
> > > In line 157, you say that "each action modifies an individual feature" -- that is not true for most causality-based CFE generation techniques, like Consequence-aware sequential counterfactual generation, Geco: Quality counterfactual explanations in real time, Amortized generation of sequential algorithmic recourses for black-box models.
> >
> > In causal-based CFE generation, constructing an SCM is essential for categorizing features into endogenous and exogenous groups. This distinction is key to identifying which features are directly influenced by the action and which are indirectly affected through their parent nodes. However, research has consistently highlighted the challenges of constructing accurate causal graphs that adequately reflect the complexities of real-world scenarios (see Barocas et al., 2020, Preserving Causal Constraints in Counterfactual Explanations for Machine Learning Classifiers, among others).
> > Our work mainly focuses on the low-level CFE generators, their CFEs, and the features the low-level actions directly affect. Our predefined actions simultaneously affect multiple features (without manually defining them), and although this doesn't address all causal issues, it addresses some of the shortcomings of manually defining which features an action affects.
> >
> > *Please let us know in case of more questions or clarifications, and thank you so much again for taking time to engage with us!*

---

> ### Comment · Reviewer_iX5Z · 2024-11-27
> **Reviewer Response to Author Rebuttal**
>
> I appreciate the time taken by the reviewers to answer my questions. My biggest concern with the approach remains still upholds because the proposed approach requires access to thousands of existing CFEs for each domain, in order to warm start their approach. I don't think that is realistic.
>
>
> > We didn’t fully understand what the reviewer meant by the term magical about this process. Section 4.1 describes the process by which we identify the hl-continuous and hl-discrete datasets
>
> That is not true, in lines 280-283, you say mention "the two types of hl-continuous actions defined by the Foods+monetary costs and Food+caloric costs, we created 4 individual -> hl-continuous datasets"? -- what does this line even mean?

---

> > ### Author Response · Authors · 2024-11-30
> >
> > We would like to thank the reviewer for their prompt reply.
> >
> > >I appreciate the time taken by the reviewers to answer my questions. My biggest concern with the approach remains still upheld because the proposed approach requires access to thousands of existing CFEs for each domain, to warm start their approach. I don't think that is realistic.
> >
> > Our proposed data-driven CFE generators address several information access constraints that decision-makers are likely to face, for example, lack of access to the classifier, classification training data, among others. Although we address several limitations in addition to information access constraints of contemporary CFE generators, one of the limitations of our work, which we mention in our limitations section, is that access to individual-CFE data may not be trivial. One solution, as we discuss at the end of Section 5, is for decision-makers to employ methods like federated learning to collaboratively train robust CFE generators under varied privacy and individual-CFE data access constraints.
> >
> > We hope the reviewer can recognize several contributions to our paper including those highlighted by other reviewers. For example, our CFE generation methods are novel, model agnostic, efficient, accurate, and offer a structured and systematic approach for fairness auditing. We tackle the challenging problem of generating higher-level actions to facilitate a more concise and interpretable communication of actionable recourse to negatively classified users. Additionally, unlike low-level CFE generators, changing more features does not reduce the explainability or interpretability of actions in high-level CFEs, and changing more features doesn’t always require more actions as it does in low-level CFEs, among other contributions.
> >
> > >That is not true, in lines 280-283, you mention "the two types of hl-continuous actions defined by the Foods+monetary costs and Food+caloric costs, we created 4 individual -> hl-continuous datasets"? -- What does this line even mean?
> >
> > Section 4.1 explains the datasets we used, and how we generated the individual-CFE datasets, with more information provided in Appendix B.1.
> >
> > The quoted sentence is contextually related to the previous paragraph (Lines 274–276).
> > The Foods dataset contains ~3k food items, each of which we associate two different costs:its monetary cost in USD and its caloric cost (see lines 274-276 and Appendix B.1.1 and B.1.3). Since some food items do not have an associated caloric cost, this results in two sets of actions, food items that have an associated monetary cost (Foods+monetary) and food items that have an associated caloric cost (Foods+caloric). For each of these two action sets, we create one individual→hl-continuous CFE dataset by solving an integer linear program (via Equation 2) given a dataset specific trained logistic regression classifier, resulting in four individual-CFE datasets. We hope that this resolves any confusion.\
> > Due to page constraints, more detailed information is added to the appendix. We also edited the PDF to make it more clear.  Please let us know if the section is still unclear.
> >
> > Lastly, we also want to mention that these distinctions between the actions were very experimentally helpful in demonstrating our methods' ability to cater to different users’ preferences and investigate fairness issues (see lines 416 to 426 and Appendix D.2.2).

---

### Official Review · Reviewer_KwJq · 2024-11-04

**Soundness:** 2
**Presentation:** 2
**Contribution:** 3
**Rating:** 6
**Confidence:** 5

**Summary:**

In this paper, the author proposes an algorithm for generating high-level counterfactual explanations (CFs). Rather than modifying individual features, the approach involves modifying groups of features simultaneously. The task is formulated as an integer linear program to obtain an optimal solution. Three different versions of the algorithm are presented for generating CFs.

**Strengths:**

Strength:
1 The topic is quite interesting.
2 The algorithm is concise and several case studies are provided.
3 The paper is well presented.

**Weaknesses:**

1 The motivation is unclear. From my understanding, they aim to define a high-level action to identify counterfactual explanations. However, they do not clearly explain why this high-level action is suitable for users to act upon. Additionally, they should clarify how these actions are identified.
2 The paper lacks a theoretical guarantee to demonstrate the efficiency of the proposed algorithm.
3 The proposed algorithms overlook the practicality of the generated counterfactual explanations (CFs).
4In Formula 2, the linear integer program (LIP) could be very time-consuming when the action space is large. They did not explain how to constrain the action space.

**Questions:**

They mention that previous methods have high computational complexity, impacting scalability. Could you clarify how your proposed algorithm addresses this issue?

---

> ### Author Response · Authors · 2024-11-27
> **Response to the reviewer KwJq**
>
> We are grateful to Reviewer KwJq for identifying the strengths of our work and for their constructive feedback. Below are the responses to their questions
>
> >However, they do not clearly explain why this high-level action is suitable for users to act upon. Additionally, they should clarify how these actions are identified.
>
> For instance, in Figure 1a, consider an individual classified as having an unhealthy waist-to-hip ratio (WHR). To help them achieve a healthy WHR classification, the low-level CFE generator provides a detailed recommendation involving 19 specific actions, each modifying a particular feature (e.g., reducing calcium intake from 309 mg to 113 mg). In contrast, the hl-continuous CFE generator provides a recommendation that is both easier to understand and easier to implement, with only two actions (take leavening agents: cream of tartar and take fish, tuna, light, canned in water, drained solids). While the hl-continuous CFE includes information on the features the actions change, the individual doesn't need to know this information, and the CFE they receive is in an implementable step that shields them from fine-grained details, making it comparatively "high-level" and easier to follow than the granular recommendations of the low-level CFE. In conclusion, high-level actions are easier for individuals to understand and implement, change multiple features, and lead to higher recourse (improvement) - see Figure 2a).
>
> We use two types of actions: Foods as an action with costs as either monetary costs, or caloric costs, i.e., Foods+monetary cost and Foods+caloric costs. In section 4.1, lines 275 to 283 and in appendix section B.1 describe the predefined actions we use.
> We edit the document to make this more explicit.
>
> >Formula 2, the linear integer program (LIP) could be time-consuming when the action space is large. They mention that previous methods have high computational complexity, impacting scalability. Could you clarify how your proposed algorithm addresses this issue?
>
> The ILPs (equations 2 and 3) provides information on the composition of the hl-continuous and hl-discrete CFEs and their generation in a non-data-driven algorithmic way. Our proposed data-driven CFE generators that rely on the instances of individual-CFE datasets are computationally cheaper because they don't require solving an NP-hard problem for each new individual. Once the  CFE generators are trained they are used to generate CFEs for several new individuals without the need for re-optimization at every generation.
>
> *Please let us know in case of more questions or clarifications, and thank you so much again for taking time to engage with us!*

---

### Official Review · Reviewer_xVRe · 2024-11-04

**Soundness:** 2
**Presentation:** 2
**Contribution:** 2
**Rating:** 5
**Confidence:** 3

**Summary:**

This paper proposes three high-level data-driven CFE generators: hl-continuous uses ILP to find the least costly set of continuous actions to change an individual's status; hl-discrete selects from discrete actions by solving a set cover problem; hl-id assigns a label to provide general recommendations without specific actions for cases where information aren’t accessible. The datasets include real-world health, some semi-synthetic and fully synthetic datasets.

**Strengths:**

- The paper provides a comprehensive analysis across diverse feature types, different dataset dimensions, and varying CFE frequencies. It also discuss fairness.
- The proposed methods are model-agnostic. So it can be adapted to different models.

**Weaknesses:**

- This paper is limited to binary classification.
- Equation (2) presumes a linear classifier, which may not represent the complexity of real-world models. This simplification could restrict the performance and generalizability of the proposed methods.
- The reliance on pre-defined parameters, such as classifier coefficients and thresholds, may reduce the flexibility and accuracy of the approach across diverse datasets.  These parameters are not directly derived from the data, potentially leading to counterfactual explanations that do not align well with actual case. The paper also does not include a sensitivity analysis of the pre-defined parameters.
- The study does not compare the proposed methods against other CFEs (only low-level CFE).

**Questions:**

I have a few questions that I’m still trying to fully understand. I would greatly appreciate any clarification you could provide on the following points:

- Why are these approaches considered "high-level"? Adjusting multiple feature combinations is not a new idea, so I’m curious about what uniquely qualifies these methods as high-level.
- How does the method handle datasets with both discrete and continuous actions?
- How are the pre-defined parameters, such as c and b, chosen? Since these parameters are not derived directly from the data, what guidelines or expertise inform their selection? Additionally, would the model’s performance benefit from dynamically adjusting these parameters based on the dataset, and is there a sensitivity analysis to evaluate how variations in bb might impact the results?

**Details Of Ethics Concerns:**

No concern.

---

> ### Author Response · Authors · 2024-11-27
> **Response to the reviewer xVRe**
>
> We thank reviewer xVRe for identifying the strengths of our work and for providing constructive feedback. Below are the responses to the questions the reviewer raised.
>
> > Equation (2) presumes a linear classifier, which may not represent the complexity of real-world models. This simplification could restrict the performance and generalizability of the proposed methods. The reliance on pre-defined parameters, such as classifier coefficients and thresholds, may reduce the flexibility and accuracy of the approach across diverse datasets. These parameters are not directly derived from the data, potentially leading to counterfactual explanations that do not align well with the actual case. The paper also does not include a sensitivity analysis of the pre-defined parameters.
>
> Using the classification training data (X, y), we train a linear model to classify the data from which we get classifier parameters. The parameters change based on the classification data choice of hyperparameter tuning. Further explanations can be found in Section 4.1 (Lines 283-284) and Appendix, Sections B.1.3.
> Furthermore, while the composition of hl-continuous and hl-discrete CFEs in the non-data-driven algorithmic cases focus specifically on binary linear classifiers, incorporating those classifier parameters into the integer linear program, our proposed data-driven CFE generators are general, model-agnostic, and scalable across diverse classification settings.
>
> > This paper is limited to binary classification.
>
> Although our experiments evaluate the performance of data-driven CFE generators primarily in binary decision scenarios (positive vs. negative), they can readily extend to other classification contexts depending on the individual-CFE datasets used to train the data-driven CFE generators.
>
> > The study does not compare the proposed methods against other CFEs (only low-level CFE).
>
> A vast majority of contemporary CFE generators, whether they are linear (including the one we directly compare with) or nonlinear, generate overly specific low-level CFEs. Those that do not overly rely on the limited classification training data (X,y) and or query access to the classifier to formulate what actions look like in the CFE (see for example,  Karimi et al., 2022; Barocas et al., 2020), which limits the extent of what individuals can do and might require extra effort (and uncounted for cost) to translate to implementable steps. Our work makes a strong case for expanding the focus beyond just classification data (e.g., classifier parameters and prediction training datasets), when automating CFE-based recourse generation.
>
>
> > The study does not compare the proposed methods against other CFEs (only low-level CFE). Why are these approaches considered "high-level"? Most current CFE generators, low-level generators, that is, generate CFEs that specifically define how much an individual should adjust their features to get a desirable outcome.  Why are these approaches considered "high-level"?
>
> Our focus was to highlight the difference between overly specific recommendations and more general ones that are easier for individuals to understand and implement. For instance, in Figure 1a, consider an individual classified as having an unhealthy waist-to-hip ratio (WHR). To help them achieve a healthy WHR classification, the low-level CFE generator provides a detailed recommendation involving 19 specific actions, each modifying a particular feature (e.g., reducing calcium intake from 309 mg to 113 mg). In contrast, the hl-continuous CFE generator suggests an easier-to-understand and implement recommendation with only two actions (take leavening agents: cream of tartar and take fish, tuna, light, canned in water, drained solids). While the hl-continuous CFE includes information on the features the actions change, the individual doesn't need to know this information, and the CFE they receive is in an implementable step which shields them from fine-grained details, making it comparatively "high-level" and easier to follow than the granular recommendations of the low-level CFE.
>
> > Adjusting multiple feature combinations is not a new idea, so I’m curious about what uniquely qualifies these methods as high-level.
>
> As the reviewer correctly points out, adjusting multiple features is not a new idea, but the difference with the proposed high-level CFEs is that the individual doesn't have to "know" which features are changed and by how much.
> Additionally, in real-world settings, recourse is not framed in specific terms as low-level CFEs (see, for example, Figure 1a).
> It is difficult to know how an individual should add the e-6 to a feature value as suggested by low-level CFEs.

---

> ### Author Response · Authors · 2024-11-27
>
> > How does the method handle datasets with both discrete and continuous actions?
>
> Each data-driven CFE generator is trained on the respective individual-CFE dataset. That is, the data-driven hl-discrete CFE generators are trained on the individual–hl-discrete CFE datasets and the data-driven hl-continuous CFE generators are trained on the individual–hl-continuous CFE datasets. The neural network models we provide in our work are experimental proofs of concept, and we show that the complexity of the CFE generator model affects the CFE generation results.
>
> >How are the pre-defined parameters, such as c and b, chosen? Since these parameters are not derived directly from the data, what guidelines or expertise inform their selection? Additionally, would the model’s performance benefit from dynamically adjusting these parameters based on the dataset, and is there a sensitivity analysis to evaluate how variations in bb might impact the results?
>
> Using the classification training data (X,y), we train a linear model to classify the data from which we get those parameters. Further explanations can be found in Section 4.1 (Lines 283-284) and Appendix, Sections B.1.3.
> When defining the composition of the hl-continuous CFEs specific to the non-data-driven algorithmic case (Equation 2), yes, those parameters of the linear classifier are crucial. We wanted to compare this setup to linear low-level CFE generators, like actionable recourse (Equation 1). Similar to the low-level CFE generators , those parameters affect not only the recourse generated but also predictions made.
>
> *Please let us know in case of more questions or clarifications, and than you again for taking time to engage with us!*

---

### Author Response · Authors · 2024-11-27
**General rebuttal summary**

We are very grateful to the reviewers for their time and effort in reviewing our work and providing valuable feedback. Their insights have been instrumental in enhancing the quality of our research.

Specifically, we thank Reviewer xVRe for acknowledging the significance of our work in addressing fairness issues and providing a comprehensive analysis across diverse feature types, dataset dimensions, and CFE frequencies. We also appreciate their recognition that our proposed methods are model-agnostic.
We are grateful to Reviewer KwJq for highlighting our work's relevance and good presentation and noting that our algorithms are concise and supported by various case studies.
We are also thankful to Reviewer iX5Z, who appreciated our approach to tackling the challenging problem of providing higher-level actions to facilitate a more concise communication of actionable recourse to negatively classified users.
Lastly, we are grateful to Reviewer cP95 for recognizing multiple strengths of our work, including the fact that changing more features does not reduce explainability/interpretability of actions in high-level CFEs as it does in the low-level CFEs, and the efficiency and accuracy of the CFE generation once the data-driven CFE generators are trained on high-quality datasets and the structured, systematic approach our method offers for fairness auditing. The reviewer also noted that generating CFEs for subgroups becomes more straightforward when predefined sets of actions are available.

Overall, the reviewers identified several strengths of our work while providing constructive suggestions for improvement. These include the need for better justification of our choice to focus on binary classification in our experiments, concise differentiation between the definitions of CFEs (constituents) and the data-driven CFE generators, and explanations of how our approach improves upon low-level, overly specific CFE generators. We address these issues in our individual responses to the reviewers and have revised the PDF within page limits to address some of the reviewer's concerns. We will further update the paper accordingly.

Please note that for the rebuttal, when referencing lines in the paper, we refer to line numbers in the initial submission, which may differ from those in the revised PDF.

Thank you once again for your thoughtful feedback and support!

---

### Meta-Review · Area_Chair_R5sn · 2024-12-23

**Metareview:**

This paper addresses counterfactual explanation (CFE) generation with a focus on (i) scalability in the underlying state space - the set of feasible states in which an individual input can exist and (ii) actionability of the CFE - can an input x actually get to the suggested x’ to cross a decision boundary, or is that path from x to x’ blocked by causal constraints etc represented as an SCM, and (iii) comprehensible CFEs.  Reviewers appreciated the motivation of the problem and its placement in the literature, as well as the coverage of the approaches given (continuous actions, discrete actions, and the hl-id setting).  Unfortunately, after a rebuttal process, all reviewers had unanswered questions, with no clear champion for the work.

**Additional Comments On Reviewer Discussion:**

We appreciate the authors' participation in the rebuttal process.  During the decision process, this AC would like to clarify that they down-weighted reviewer iX5Z's overly-negative take on the paper -- some concerns raised were certainly valid, and some remain unanswered, but (having also personally looked at this paper in some depth) the work is clearly not a "1 - reject" -- and we encourage the authors not to anchor on that.  That said, the work appears to need another round of revision prior to appearing, and this AC encourages the authors to take their rebuttal and subsequent discussion into account in a revision.

---

### Decision · Program_Chairs · 2025-01-22

Reject